# Human cytomegalovirus IE2 drives transcription initiation from a select subset of late infection viral promoters by host RNA polymerase II

**Ming Li**[1,2], **Christopher B. Ball**[2], **Geoffrey Collins**[2], **Qiaolin Hu**[1], **Donal S. Luse**[3], **David H. Price**[2], **Jeffery L. Meier**[1]*

**1** Departments of Internal Medicine and Epidemiology, University of Iowa and Iowa City Veterans Affairs Health Care System, Iowa City, IA, United States of America, **2** Department of Biochemistry, University of Iowa, Iowa City, IA , United States of America, **3** Department of Cardiovascular and Metabolic Sciences, Lerner Research Institute, Cleveland Clinic, Cleveland, OH, United States of America

* Jeffery-meier@uiowa.edu

**Data Availability Statement:** URLs are within the manuscript to analyze relevant data. The Github links are https://github.com/meierjl/Towne, https://github.com/meierjl/Towne, and https://github.com/

## Abstract

Herpesvirus late promoters activate gene expression after viral DNA synthesis has begun. Alphaherpesviruses utilize a viral immediate-early protein to do this, whereas beta- and gammaherpesviruses primarily use a 6-member set of viral late-acting transcription factors (LTF) that are drawn to a TATT sequence in the late promoter. The betaherpesvirus, human cytomegalovirus (HCMV), produces three immediate-early 2 protein isoforms, IE2-86, IE2-60, IE2-40, late in infection, but whether they activate late viral promoters is unknown. Here, we quickly degrade the IE2 proteins in late infection using dTag methodology and analyze effects on transcription using customized PRO-Seq and computational methods combined with multiple validation methods. We discover that the IE2 proteins selectively drive RNA Pol II transcription initiation at a subset of viral early-late and late promoters common to different HCMV strains, but do not substantially affect Pol II transcription of the 9,942 expressed host genes. Most of the IE2-activated viral late infection promoters lack the TATT sequence bound by the HCMV UL87-encoded LTF. The HCMV TATT-binding protein is not mechanistically involved in late RNA expression from the IE2-activated TATT-less UL83 (pp65) promoter, as it is for the TATT-containing UL82 (pp71) promoter. While antecedent viral DNA synthesis is necessary for transcription from the late infection viral promoters, continued viral DNA synthesis is unnecessary. We conclude that in late infection the IE2 proteins target a distinct subset of HCMV early-late and late promoters for transcription initiation by RNA Pol II. Commencement of viral DNA replication renders the HCMV genome late promoters susceptible to late-acting viral transcription factors.

meierjl/truQuant. The GEO datasets, GSE139114, are available at https://www.ncbi.nlm.nih.gov/geo/query/acc.cgi?acc=GSE139114.

**Funding:** The funders listed below had no role in study design, data collection and analysis, decision to publish, or preparation of the manuscript. National Institute of General Medical Science (R35-GM126908 to DHP.; R01-GM121428 to DSL), National Institute of Allergy and Infectious Diseases R21-AI130453 to DHP and JLM), University of Iowa Carver College of Medicine Collaborative grant (to DHP and JLM), and the Department of Veterans Affairs (Merit award I01 BX004434 to JLM).

**Competing interests:** All authors declared that no competing interests exist.

## Author summary

The herpesvirus subfamilies differ in the viral proteins used in generating the cascade of viral immediate-early, early, early-late, or late gene transcription. With the application of advanced technologies, we discovered that the betaherpesvirus, human cytomegalovirus, has evolved strategies analogous to those used by both alpha- and gammaherpesviruses to bring about RNA Pol II transcription from its late infection promoters. Like alphaherpesviruses, human cytomegalovirus purposes a pivotal immediate-early viral transcription factor to initiate transcription from early, early-late, and late viral promoters. However, the cytomegalovirus transcription factor only targets a select set of viral early-late and late promoters without appreciably affecting host promoters at late times. Most of these late infection viral promoters are structurally and mechanistically different from promoters activated by the 6-member viral transcription factor complex that is analogous to the transcription factor complex utilized by gammaherpesviruses. Human cytomegalovirus genome amplification must first take place, but need not continue, to enable the two different mechanisms of late viral promoter activation.

## Introduction

Over half of humankind is infected by human cytomegalovirus (HCMV), the prototype of the beta-herpesvirus subfamily. While HCMV usually does not harm its only host, immature or impaired cellular immunity allows unrestrained HCMV replication in organ tissues that causes life-threatening disease. Today's antiviral drugs are potentially toxic and not always effective. In the early days of the AIDS epidemic the first anti-sense nucleic acid therapeutic was FDA-approved for the local treatment of CMV retinitis. This phosphorothioate oligodeoxynucleotide was designed to target the mRNA for the HCMV immediate-early IE2 protein, based on results of prior studies establishing IE2's pivotal role in HCMV replication [1–3]. IE2 is expressed immediately upon infection of cell types permitting viral replication. The HCMV major immediate-early promoter (MIEP) and enhancer govern this expression by directing host polymerase II (Pol II) and associated general transcription factors (GTF) to produce poly-adenylated pre-mRNA that spans exons 1 through 5 of the major immediate-early (MIE) locus. The differential splicing of this message yields IE1 and IE2 mRNAs encoding IE1-72 kDa (IE1-72) and IE2-86 kDa (IE2-86) proteins, respectively. The mature IE1 and IE2 mRNAs share exons 1–3 that code for the first 85 amino acids in both IE1-72 and IE2-86. Exon 5 codes for the additional 494 amino acids in IE2-86, whereas exon 4 codes for the carboxyl portion of IE1-72. IE2-86 transactivates HCMV early kinetic-class promoters in plasmid reporter constructs [4–6] and in the natural context of the HCMV genome in infected cells [7, 8].

Four different paradigms that are not mutually exclusive have been proposed to explain how IE2-86 activates transcription from viral early promoters: 1) IE2-86 acts through cis-regulatory elements (CRE) by binding to host transcription factors or adapter proteins that occupy the CRE [9]. 2) IE2-86 acts on host GTFs by binding to a GTF and possibly functioning like a TATA-associated factor [9–11]. 3) IE2-86 promotes transcription by binding directly to DNA in the vicinity of the viral early promoter, with preference for specific DNA sequences [12, 13]. 4) IE2-86 counters the repressive effects of chromatin assembly [14, 15] and intrinsic immunity [16] by interacting with components of these systems. In contrast to IE2-86's propensity to activate promoters, rising levels of IE2-86 feedback to decrease transcription from the MIEP [17, 18]. This repression results from an IE2-86 dimer binding to a 14-bp cis repression

sequence (*crs*) positioned between the MIEP TATA box and initiator sequence (Inr) to thereby block transcription initiation [19–21].

All herpesvirus family members apply a strategy that connects onset of viral DNA replication with onset of transcription from late viral promoters. Alphaherpesviruses do this with the active contribution of a viral immediate-early protein. For example, the herpes simplex virus (HSV) immediate early protein ICP4 activates transcription from both early and late viral promoters. Onset of HSV DNA replication permanently renders the viral genome accessible to host GTF, and ICP4 is continuously required for the Pol II occupancy of viral late promoters [22]. ICP4 drives host preinitiation complex formation at HSV late promoters that may only consist of a TATA box and an Inr sequence [23, 24]. Beta- and gammaherpesviruses have evolved viral late promoters with a TATT sequence positioned upstream of the transcription start site (TSS), replacing the canonical TATA, and a viral late-acting transcription factor (LTF) that binds to this sequence. The viral TATT-binding protein (vTBP) is part of a 6-component assembly containing 5 additional different viral LTFs. The HCMV vTBP, encoded by UL87, and its interacting LTF partners (UL49, UL79, UL91, UL92, and UL95) bear little structural resemblance to their gammaherpesvirus counterparts [25–27]. The onset of viral DNA replication and LTFs are required to drive RNA expression from the TATT-containing late viral UL44 promoter [25], as well as the expression of several other viral late RNAs [28–30]. The viral late IE2-40 and IE2-60 transcripts arise from TATT-containing promoters located at the 5' end of exon 5 for IE2-86 [3, 31–33]. These transcripts are translated in frame with the carboxy portion of IE2-86 to produce the late IE2-40 and IE2-60 proteins [3, 34].

The purpose of having three IE2 protein isoforms abundantly present throughout late infection is poorly understood. These proteins appear to have somewhat different yet overlapping functions. IE2-40 functions like IE2-86 in binding to the *crs* to dampen transcription from the MIEP [32, 35, 36]. In transient reporter assays, IE2-40 transactivates select heterologous promoter types when expressed with IE1, but does not transactivate the HCMV early promoters that respond to IE2-86 [35]. In trans-complementation studies, the ectopic expression of IE2-86 and IE2-40 together was better than IE2-86 alone in rescuing an HCMV recombinant lacking exon 5 that codes for all IE2 isoforms [37]. Eliminating the expression of both IE2-60 and IE2-40 by mutation of their promoters lowers production of infectious viral progeny [38]. The mutations also decrease UL83 RNA level, as measured by quantitative PCR [32], and lower UL84 protein amount through a post-transcriptional mechanism involving IE2-40 [39]. Because HCMV genes commonly have multiple promoters that function in late infection [33] and not all TSSs produce stable nascent RNAs [40, 41], methods that only analyze mRNA sequences are not sufficient to fully elucidate if and how the IE2 proteins effect viral transcription in late infection.

In this report, we adapt a new method of targeted degradation [42] to deplete all IE2 protein isoforms over a 6 h-timeframe in late infection. A customized PRO-Seq [33, 40] methodology is applied to quantitatively measure nascent RNA Pol II transcripts and profile the amount of transcription start-site utilization, promoter proximal pausing, and productive elongation across viral and host genomes. This combined approach coupled with multiple validation methods uncovered a subset of late infection viral promoters that rely on the IE2 and antecedent viral DNA synthesis for transcription initiation. Host promoters are not appreciably affected by the depletion of IE2. The IE2-activated early-late and late viral promoters commonly lack a vTBP-recognition sequence or the canonical TATA sequence and do not universally involve vTBP for transcription. Viral promoter transcription initiation by IE2 or vTBP is not dependent on continuation of viral DNA replication fork progression. The start of viral DNA replication enduringly renders viral genomes receptive to the transcription initiation mediated by two different mechanisms of viral promoter activation, with one mechanism

resembling that used by alphaherpesviruses and the other mechanism resembling that used by gammaherpesviruses.

## Results

### Targeted degradation of the IE2 protein isoforms in late infection

A recently developed targeted degradation technology [42] was adapted to rapidly deplete all IE2 protein isoforms on-demand as a means to elucidate the role of these proteins in late infection. The degradation tag molecule (dTag) joins the FKBP12$^{F36V}$-tagged protein with the cereblon E3 ubiquitin ligase complex leading to cereblon-mediated degradation by the host proteasome (**Fig 1A**). The dTag molecule has the cereblon ligand, thalidomide, appended to a synthetic FKBP12$^{F36V}$-directed ligand, AP18671, which does not bind to wild-type FKBP12 [43]. In the HCMV Towne strain genome, FKBP12$^{F36V}$ was fused in-frame to the carboxyl-terminal ends of the IE2 proteins to produce the HCMV IE2F virus (**Fig 1B, S1 Fig**). During infection of human foreskin fibroblasts (HFF), the tagged IE2F-86, IE2F-60, and IE2F-40 proteins were expressed at a rate and in amounts similar to that of the untagged IE2 counterparts expressed by the wildtype HCMV IE2 virus (**Fig 1B**). The viral late IE2F-40, IE2F-60, and pp28 proteins became abundant beginning at 72 h post-infection (pi). The wildtype IE2 and tagged IE2F viruses also produced equivalent amounts of new viral genomes over 96 h (**S2A Fig**), indicating equivalence in number of viral DNA templates that could potentially be transcribed in late infection. Based on this information, HFF infected with HCMV IE2 versus IE2F were exposed to the dTag degrader (200 nM) or vehicle control at 90–96 h pi and the outcome analyzed by western blot at 96 h pi. The dTag treatment markedly lowered the amounts of all three IE2F protein isoforms, but not the amounts of viral pp28 or untagged IE2 proteins (**Fig 1C**). While maximal depletion of IE2F proteins was mostly achieved as early as 2 h after 200 nM dTag treatment (**S2B Fig**), we opted to treat for 6 h to also allow time for change in steady-state level of RNA. The short time frame of 6-h dTag treatment at 90–96 h pi had not lowered the amount of viral DNA or infectious particle produced (**S2C Fig**). Select viral RNAs were quantified by real-time PCR (qPCR) in a set of infections carried out in parallel to those assessed by western blot in Fig 1C. The results show that dTag-mediated IE2F depletion increased levels of spliced IE1-72 and IE2-86 RNAs by 115–366% (**Fig 1D**), consistent with the reduction in negative autoregulation of the MIE promoter imparted by IE2-86 and IE2-40. In contrast, the level of spliced RNA from the neighboring upstream UL128 gene had modestly decreased by 54% and RNA levels of viral late pp28 and early-late GFP had not appreciably changed. This pattern of diametrical changes in RNA levels for the same set of genes was reproduced in two additional independent experiments carried out on different donor HFF cells. RNA levels from the same genes of wildtype HCMV IE2 were not significantly changed by dTag treatment.

### PRO-Seq provides mechanistic insight into IE2's control of viral transcription in late infection

A customized PRO-Seq protocol was applied to determine if targeted degradation of IE2F in late infection affects viral transcription. Recent refinements to the PRO-Seq method [44] has improved the accuracy in measuring the frequency with which Pol II and its nascent transcript is located at any nucleotide position along either strand of the HCMV genome [33]. Adding the P-TEFb inhibitor flavopiridol (Flavo) for the last 1 h of the infection in a parallel set of infections allows PRO-Seq to determine precisely where on the viral genome transcription is initiating, the extent of promoter-proximal Pol II pausing, and the amount of productive

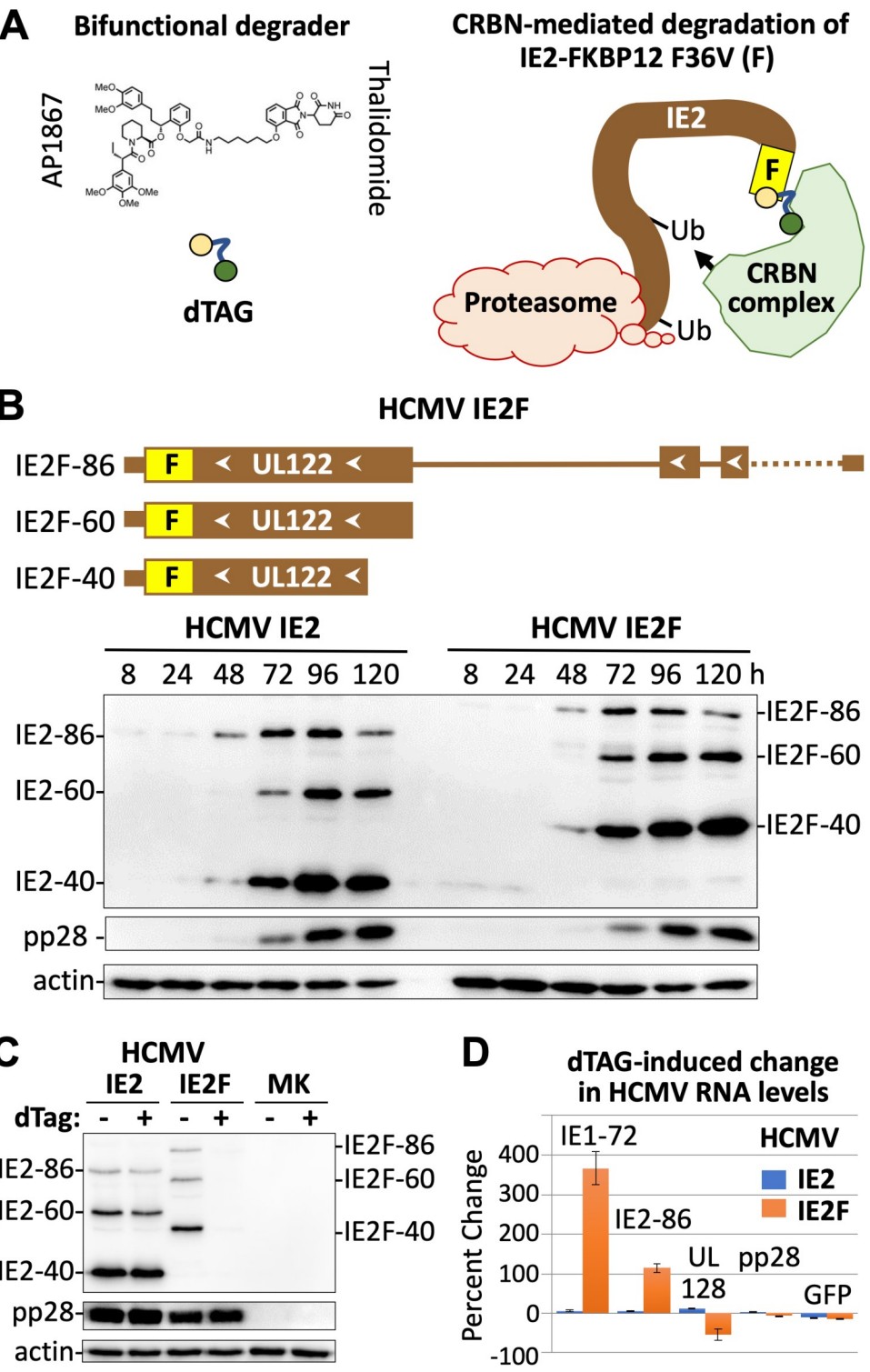

**Fig 1. Targeted degradation of IE2 in late infection–proof of principle. (A)** Schematic depiction of the targeted degradation strategy using dTag to target an IE2 protein tagged with FKBP12$^{F36V}$ (F) for degradation by the host cereblon (CRBN)-ubiquitination-proteasome pathway. **(B)** FKBP12$^{F36V}$ was fused in-frame to carboxyl ends of IE2-86, IE2-60, and IE2-40 in the HCMV Towne recombinant, named HCMV IE2F. Western blot shows amount of tagged IE2-86F, IE2-60F, and IE2-40F proteins expressed in HCMV IE2F-infected HFF over time, compared to untagged IE2 expressed from HCMV IE2 at multiplicity of infection (MOI) of 1 infectious unit per cell. **(C)** Western blot shows change in IE2F levels at 96 h pi resulting from 200 nM dTag (vs. vehicle) treatment from 90–96 h pi, as compared to

untagged pp28 or IE2 (MOI of 1). (**D**) RT-qPCR results reveal that dTag treatment of IE2F-infected HFF for 90–96 h pi produces diametrically opposed changes in RNA levels for specific viral genes whose RNAs do not change in the untagged IE2-infected HFF (MOI of 1).

transcription elongation that is taking place. Making such determinations using the PRO-Seq plus Flavo approach has been verified by PRO-Cap that reports on 5'-end-capped transcripts [33]. We have assembled our PRO-Seq data in an HCMV Towne track hub made available on GitHub: https://github.com/meierjl/Towne. The hub can be loaded in the UCSC Genome Browser to view the PRO-Seq results. A zoomed-in browser view of the MIE gene region reveals a noticeable pattern of change in transcription resulting from IE2F depletion that parallels the changes in RNA levels measured by qPCR (**Fig 2A**). The large increase in number of non-duplicative paired-end reads mapping to the MIEP and covering the MIE genes reflects an increase in transcription, whereas the marked decrease in number of reads spanning the neighboring upstream gene blocks, UL128 and UL130, indicates less transcription. Remarkably, the IE2F depletion also abrogated transcription from the recently described promoter in the proximal enhancer for the MIE promoter [33]. Treatment of wildtype HCMV IE2 with dTag had not changed the read coverage across the MIE region, further validating the specificity of the targeted degradation strategy. In viewing the PRO-Seq plus Flavo results, the enormous increase in pileup of reads at the MIEP TSS and into the promoter-proximal pause zone of Pol II stands out in the IE2F depletion track (**Fig 2B**). This finding is concordant with an increase in level of transcription from the MIEP TSS. In contrast, the stacks of reads distinguishing other promoters in the UL128-UL130 region nearly disappeared as a consequence of disposing of IE2F. This reduction in reads at the promoter TSS and in the promoter-proximal pause zone is indicative of a decrease in transcription initiation. The MIEP TSS and the TSS in MIE proximal enhancer that are spaced 125-bp apart are prime examples where IE2 has opposite effects on levels of transcription initiation.

We next determined if the IE2-activated promoters fit the conventional definition of a herpesvirus late promoter that only becomes active after viral DNA replication has taken place. For these studies, we used the HCMV TB40/E strain to verify that the viral promoters of interest are not a quirk of an attenuated laboratory HCMV strain. The HCMV TB40/E strain closely resembles a clinical strain and has blocks of genes (UL133-144 and UL148A-UL150) the Towne strain lacks. To normalize PRO-Seq reads for the different treatment conditions, spike-in Sf21 moth nuclei controls were added to infected HFF nuclei. The alignments of normalized PRO-Seq reads across the TB40/E genome by treatment condition are viewable by accessing the TB40/E browser track hub: https://github.com/meierjl/TB40E. Focusing on the promoter in the MIE proximal enhancer in **Fig 3** as the exemplar reveals a prominent peak of reads at 72 h pi that reflects a relatively high frequency of promoter proximal Pol II occupancy. The peak was nearly eliminated by the viral DNA synthesis inhibitor, PFA, when it was present throughout infection. The peak was not evident before viral DNA synthesis at 24 h pi. These combined results indicate that the MIE proximal enhancer promoter operates like a classical viral late promoter.

## Validation of PRO-Seq measurements in determining transcription effects of IE2 loss

The reproducibility of the HCMV Towne IE2F PRO-Seq findings was first demonstrated using the HCMV TB40/E strain having the same IE2F configuration (**S1 Fig**). Testing of HCMV TB40/E IE2F was carried out at 72 h pi instead of 96 h pi to confirm that the findings extend to earlier time points in late infection. To more accurately measure the degree of

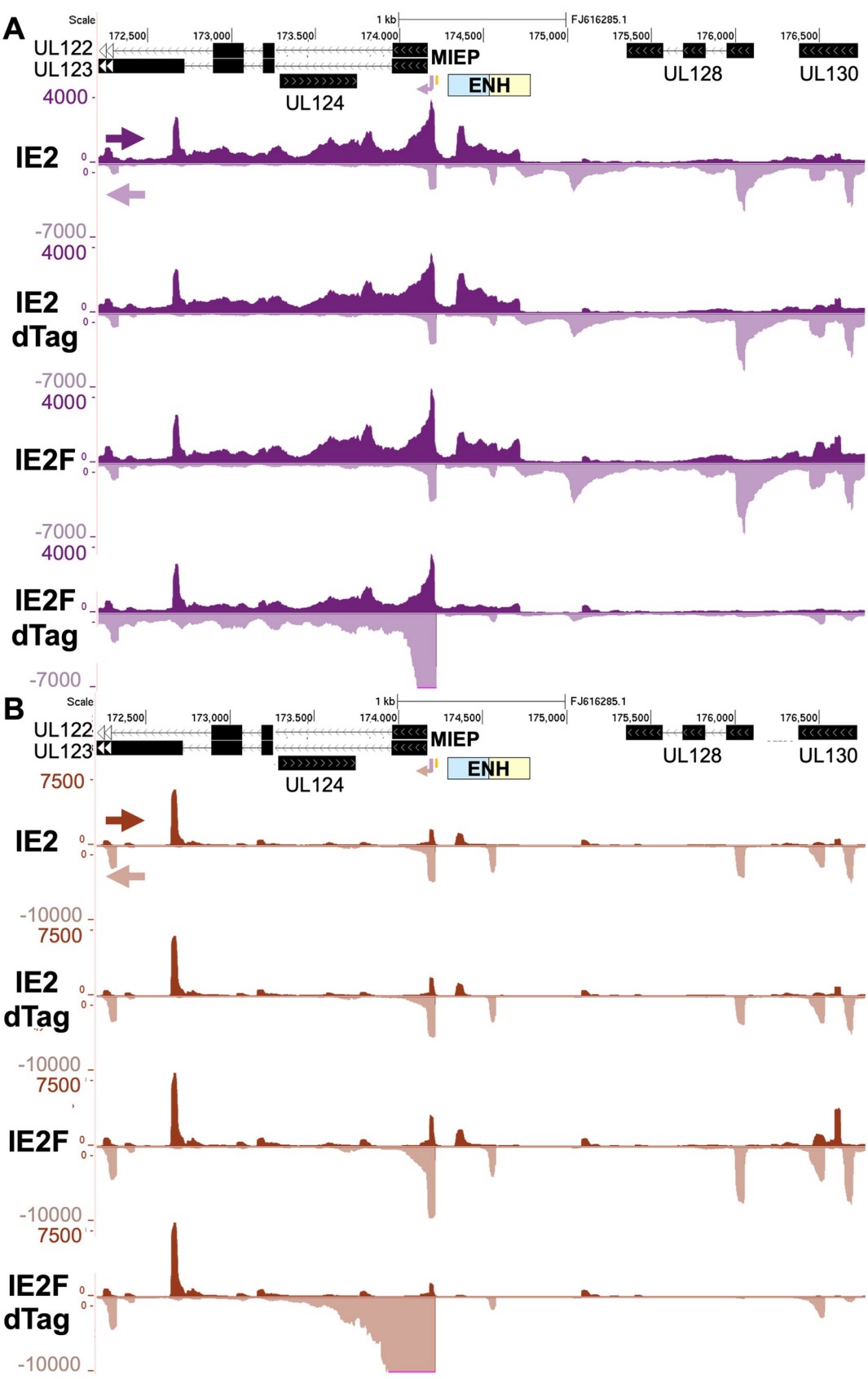

**Fig 2. Effect of IE2F depletion on transcription in the MIE region. (A and B)** Browser views of PRO-Seq results for HFF infected with untagged IE2 (IE2) or F-tagged IE2 (IE2F) HCMV Towne treated with vehicle or dTag (200 nM) from 90–96 h pi. Flavopiridol (1 μM, Flavo) was added for the final 1 h of infection. PRO-Seq reads were aligned to the HCMV Towne genome (GenBank accession number FJ16285.1). Arrows denote direction of transcription. The MIEP enhancer is composed of dissimilar proximal (blue) and distal (beige) segments. The MIEP *crs* (vertical orange bar) is located immediately upstream of the TSS.

change between the different treatment conditions, the same number of moth cells were added to each of the infected HFF samples immediately prior to nuclei preparation to provide the internal reference for normalizing PRO-Seq results. The PRO-Seq read alignments to the TB40/E genome for two sets of independent experiments are viewable by accessing the TB40/E browser track hub. At 72 h pi, between 5.6 to 6.4 million non-duplicative reads aligned to the HCMV TB40/E genome and the number of reads decreased by 5–7% as a result of the 6 h-treatment with 200 nM dTag (S1 Table). We have opted to spotlight the UL83 gene for additional scrutiny because it codes for the most abundant virion protein, greatly ramps up production of polyadenylated transcripts after viral DNA replication has started [45], and is driven by a promoter that is highly affected by the 6 h-depletion of IE2F proteins in late

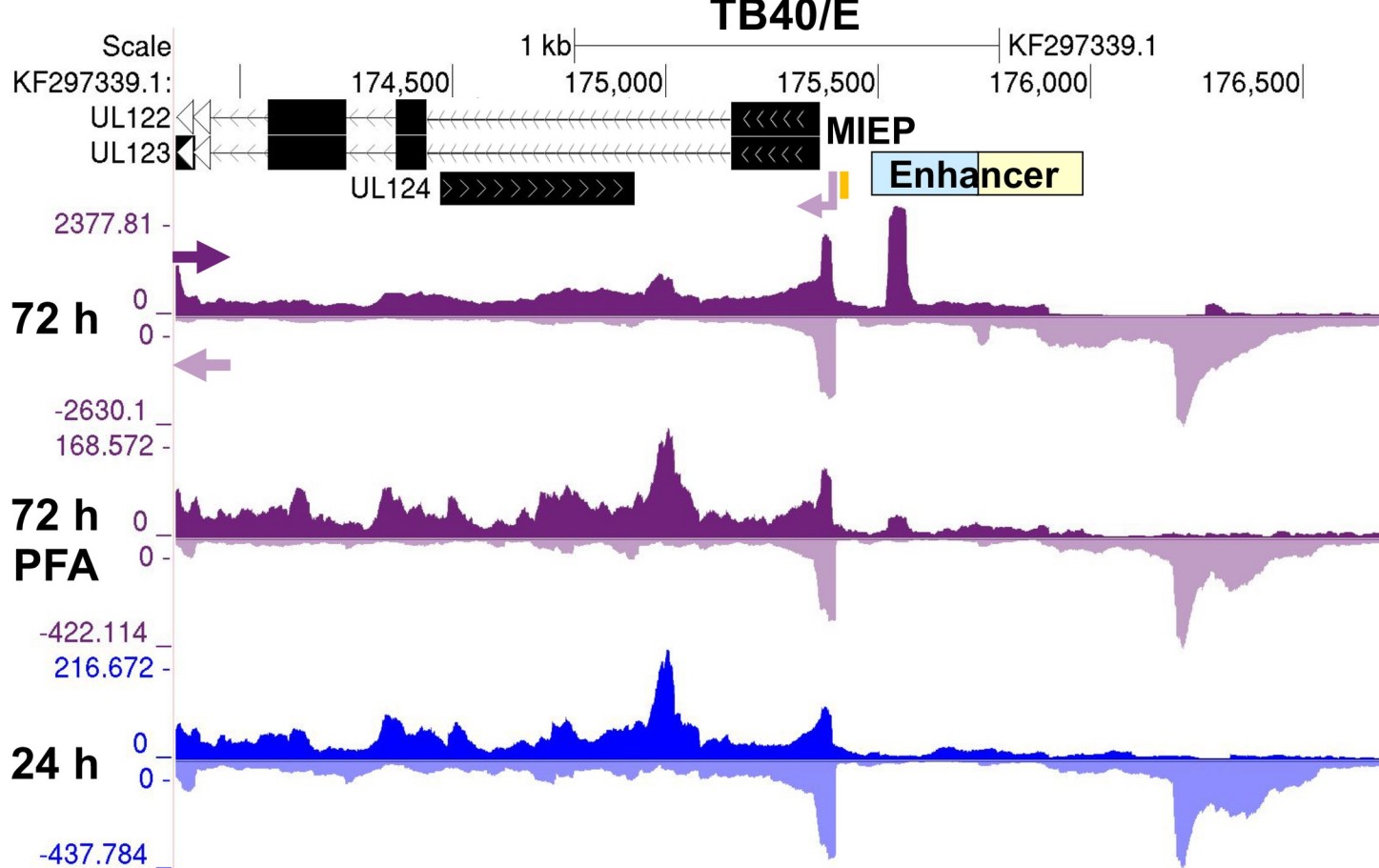

**Fig 3. Effect of preempting viral DNA synthesis on transcription in the MIE region.** Browser views of transcription from the proximal MIE enhancer promoter from PRO-Seq results for HCMV TB40/E infections at 24 and 72 h pi. PFA (400 μg/mL) was present throughout a 72 h-infection. Viral reads were normalized to reads from spike-in moth nuclei and aligned to HCMV TB40/E genome (GenBank accession number KF297339.1). Arrows denote direction of transcription. The MIEP enhancer is composed of dissimilar proximal (blue) and distal (beige) segments. The MIEP *crs* (vertical orange bar) is located immediately upstream of the TSS.

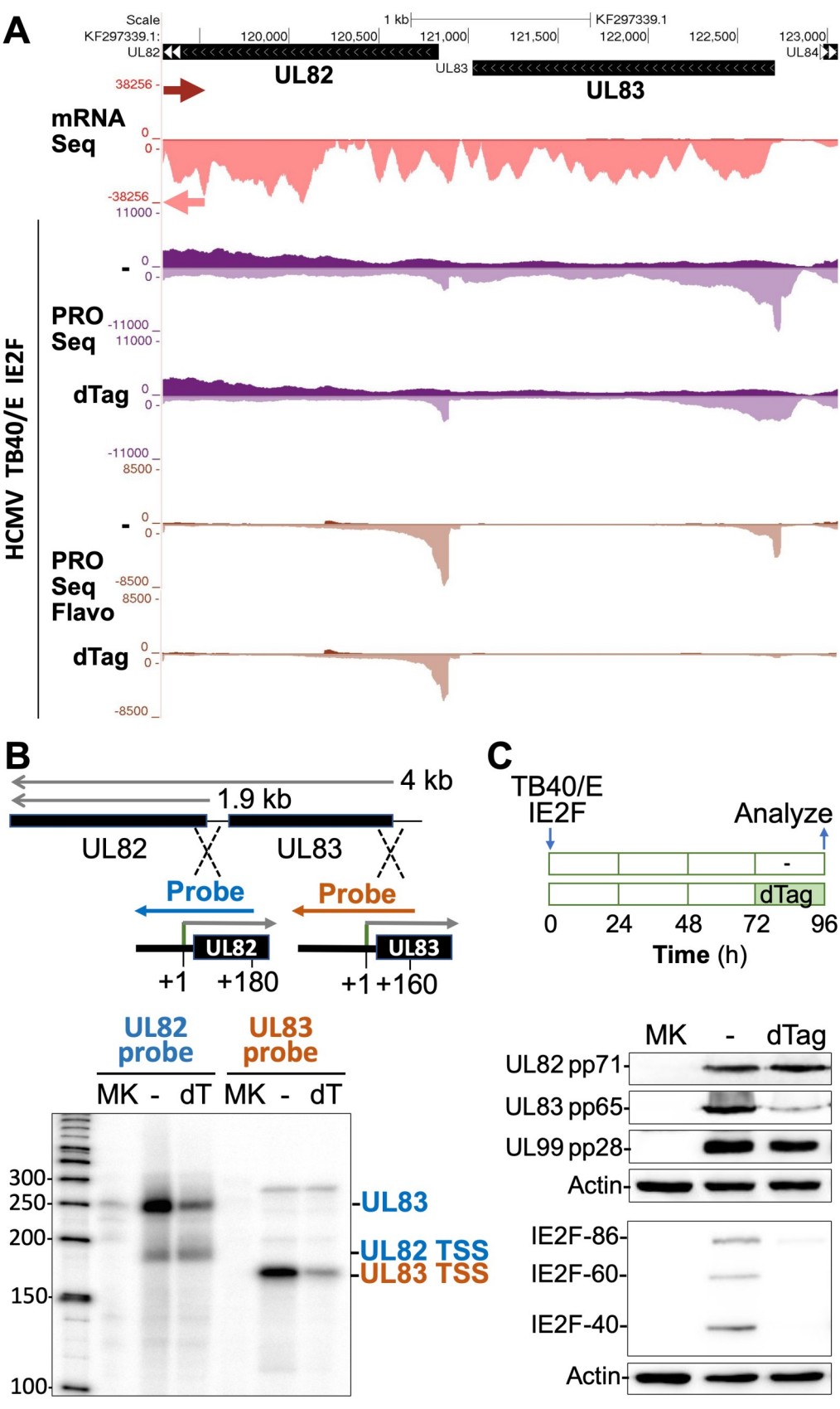

**Fig 4. IE2 drives late UL83 transcription initiation for pp65 expression. (A)** Browser views over the HCMV UL82 and UL83 genes. HCMV Merlin mRNA-Seq reads from infected fibroblast at 72 h pi (GSE41605) (47) were aligned to the HCMV TB40/E genome (GenBank accession number KF297339.1). PRO-Seq was applied at 72 h pi to HFF infected with HCMV TB40/E IE2F plus or minus 6h-dTag (200 nM) for 66–72 hpi. Flavo was added for the final 1 h of infection. Viral reads were normalized to reads from spike-in moth nuclei and aligned to HCMV TB40/E genome. Arrows denote direction of transcription. **(B)** Depiction of the $P^{32}$-labeled UL82 and UL83 riboprobes fashioned to measure levels of RNAs arising from TB40/E UL82 TSS (180 nt) and UL83 TSS (160 nt), respectively. The UL82 riboprobe also detects the UL83 4.0 kb-mRNA spanning UL82 (46). RPA was performed at 72 h pi plus or minus 6h-dTag (200 nM) for 66–72 hpi and analyzed with a phosphoimager. Mock infection, MK. **(C)** At 72 h pi, 200 nM dTag was added or not for 24 h and then analyzed, as schematically depicted. The indicated proteins in whole cell extracts were assessed by western blot.

infection. As shown in **Fig 4A,** treatment with dTag for 6 h resulted in a 75–80% reduction in number of PRO-Seq reads aligning to the UL83 gene body, compared to the treatment with vehicle alone. In the Flavo-treated group, the pileup of 4,480 PRO-Seq reads at the UL83 TSS decreased by ~90% as a result of the dTag treatment, while the UL82 TSS read-count changed negligibly. We next determined if these findings closely paralleled expression levels of UL83 gene products measured by other methods. UL83 is known to produce a 4-kb mRNA that increases greatly in amount in late infection [45] and terminates downstream of UL82 along with the 1.9-kb mRNA produced by the late UL82 promoter [46] (**Fig 4B**). RNAse protection assay (RPA) with strand specific UL82 and UL83 riboprobes straddling the UL82 and UL83 TSSs, respectively, was applied to measure the amount of RNA arising from these viral promoters after dTag treatment for 6 h at 66–72 h pi (**Fig 4B**). As expected, the UL82 riboprobe measurements revealed that IE2F depletion does not decrease the amount of RNA arising from the UL82 promoter but does decrease the amount of UL83 RNA crossing the UL82 gene body. This latter finding corresponds to the UL83 riboprobe result indicating that IE2F depletion produces ~80% decrease in amount of RNA arising from the UL83 promoter. We then determined if change in UL83 RNA level translated into change in pp65 protein level. To allow time for change in pp65 level, the duration of dTag treatment was lengthened to 24 h during the interval of 72–96 h pi (**Fig 4C**). As anticipated, the dTag treatment markedly diminished pp65 abundance, while levels of the late UL82 and UL99 gene products, pp71 and pp28, respectively, held constant.

## Characteristics of IE2-dependent promoters

Herpesvirus 'true' late genes are expressed only after viral DNA replication, whereas early-late genes are expressed at much lower level before viral DNA replication than they are after viral DNA replication. In addressing the extent to which IE2 activates these two types of viral late promoters, we first mapped all active viral TSSs in late infection by identification of a corresponding transcription start region (TSR) composed of >200 nonduplicative 5'-end reads beginning within a 20-base interval. PRO-Seq datasets from Flavo-treated infections were used to locate these viral TSRs across the TB40/E and Towne genomes at 72 and 96 h pi, respectively. We next determined the effect of IE2F depletion on number of reads at each viral TSR using the spike-in normalized PRO-Seq plus Flavo datasets for dTag-treated versus control (CTRL)-treated TB40/E IE2F and Towne IE2F infections at 72 and 96 h, respectively. Depletion of IE2F caused >2-fold decrease in number of reads among 7.6–12% of eligible TSRs, indicating that a subset of viral TSSs depend on IE2 for activation in late infection (**S3 Fig**). The opposite outcome of >2-fold increase in read number occurred among 1–1.2% of eligible TSRs. This implies that the MIEP TSS is not the only TSS that is repressed by IE2. The degree to which inhibition of viral DNA replication from the start affects TSR development was

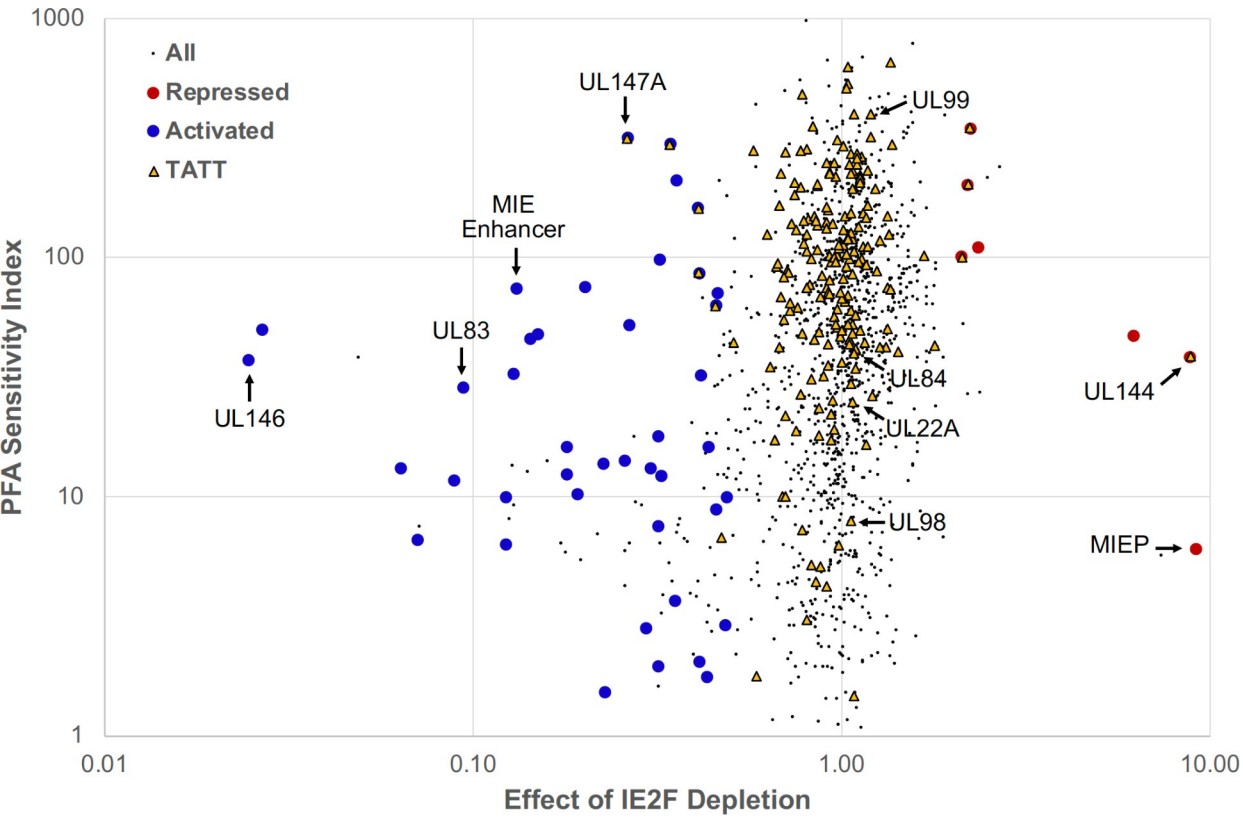

**Fig 5. PSI score versus effect of IE2F depletion on TSR strength.** Each TSR is indicated by a small black dot. Repressed and activated TSRs increase and decrease in strength, respectively, when IE2F is depleted. Repressed TSRs containing a dTag / CTRL ratio greater than 2 are red. Activated TSRs with ratios less than 0.5 are blue. TSRs with an upstream TATT sequence are shown as yellow triangles. TSRs immediately downstream (within ~200 bp) from major TSRs that are due to 5´ ends generated by hydrolysis and not actual transcription initiation were removed from the repressed and activated lists. Such TSRs still appear in the plot as black dots but are not identified as activated or repressed. Arrows point to TSRs for the indicated viral promoters.

determined by quantitative analysis of normalized PRO-Seq datasets for TB40/E infections at 72 h carried out in the presence versus absence of the PFA inhibitor of viral DNA replication. The No PFA/PFA ratio of reads, labeled as the PFA sensitivity index (PSI), was computed for each TSR. Registering a high PSI value on the PSI scale reflects a high degree of dependency on viral DNA replication for TSR read production. The distribution of the PSI values in relation to amount of change in TSR reads resulting from TB40E IE2F depletion (dTag/CTRL TSR reads) is displayed by scatter plot for all eligible TSRs (black dots) (**Fig 5**). TSRs with a TATT sequence positioned at -37 to -30 upstream of the maximum TSS are marked with yellow triangles. A small minority of eligible TSRs occurring immediately downstream from major TSRs may represent smaller piles of 5'-ends that are generated by RNA hydrolysis during library construction [33]. They are not part of the 42 different IE2-activated TSRs marked with large blue dots or the 7 different IE2-repressed TSRs marked with large red dots. These marked TSRs are constants in other HCMV PRO-Seq and PRO-Cap datasets from HFF infections at late times [33]. For the IE2-activated TSRs, PSIs ranged widely from 1.5 to 318 (**Table 1**, **S1 Data**). The IE2-activated MIE proximal enhancer TSR provides a frame of reference for lateness in transcription because it is barely evident before viral DNA replication at 24 h pi (**Fig 3**). Its PSI of 75.5 scores in the lateness range of PSIs for the bulk of IE2-unaffected TSRs having an upstream TATT and is exceeded by the PSIs of 7 other IE2-activated TSRs.

**Table 1. IE2-activated HCMV transcription start sites.**

| Promoter Location | TSS Position | Fold-Change | PSI | TATW |
|---|---|---|---|---|
| UL146 | 181708L | 40.5 | 37.4 | None |
| ORFL274W | 177771R | 37.2 | 50.8 | None |
| UL130 AS | 177661R | 15.6 | 13.2 | None |
| RL11 AS | 9889L | 14.1 | 6.7 | None |
| UL130 I1 | 177346L | 11.3 | 11.8 | None |
| UL83 | 122740L | 10.6 | 28.8 | None |
| ORFL206W | 122981R | 8.1 | 10.0 | None |
| US14 | 211621L | 8.1 | 6.4 | None |
| UL9 I | 18035R | 7.7 | 33.0 | None |
| MIE Enhancer P1 | 175525R | 7.6 | 75.5 | None |
| ORFL271C.iORF1 | 177239L | 7.0 | 46.3 | None |
| UL33 | 43791R | 6.6 | 48.5 | None |
| ORFL9C | 5850L | 5.6 | 12.5 | None |
| UL133 I1 | 190424L | 5.6 | 16.2 | None |
| UL130 | 178103L | 5.2 | 10.4 | None |
| UL130 I2 | 177951L | 5.0 | 75.8 | None |
| RL10 AS | 9212L | 4.4 | 1.5 | None |
| UL139 | 187313L | 4.4 | 13.9 | TATAATGTGCAC |
| ORFL33W | 13799R | 3.9 | 14.4 | TATAAGGATCGC |
| ORFL66W.iORF1 | 29735R | 3.8 | 52.9 | None |
| UL147A | 181119L | 3.8 | 318.1 | TATTTGACCGCA |
| UL148D | 192302R | 3.4 | 2.8 | TATAAAGCCACC |
| ORFL167W | 104747R | 3.3 | 13.3 | None |
| ORFL39C | 16337L | 3.1 | 7.6 | None |
| oriLyt | 92858L | 3.1 | 18.0 | None |
| ORFL191C | 113119L | 3.1 | 2.0 | None |
| UL133 I2 | 190659L | 3.1 | 12.3 | None |
| US28 AS | 228607L | 3.1 | 98.4 | None |
| UL6 AS | 16185L | 2.9 | 300.6 | TATATTAACAAA |
| RNA5.0 I | 158019L | 2.8 | 213.3 | None |
| UL114 AS | 164579R | 2.8 | 3.7 | TATATATATTGT |
| RL12 | 10013R | 2.4 | 2.1 | None |
| ORFL207W | 123407R | 2.4 | 32.3 | None |
| UL142 AS | 184469R | 2.4 | 86.9 | TATTAATAAAGA |
| ORFL310W | 187229R | 2.4 | 162.4 | TATTACTATATA |
| RNA1.2 | 8105L | 2.3 | 16.3 | None |
| UL35 I | 48527R | 2.3 | 1.8 | None |
| UL52 AS | 77313L | 2.2 | 8.9 | TATATAGTAATT |
| UL128 | 177713L | 2.2 | 63.7 | TATTAACGGGTC |
| UL146 AS | 181285R | 2.2 | 72.2 | TATATACGTGAT |
| ORFL18W | 7908R | 2.1 | 10.0 | None |
| UL103 | 151768L | 2.1 | 3.0 | None |

IE2-mediated effect on viral TSRs with >200 reads determined from normalized Flavo-treated HCMV TB40/E IE2F datasets for 6h-treatment with vehicle vs dTag at 72 h pi. Maximum TSS position in annotated TB40/E genome (GenBank accession No. KF297339.1) with predicted promoter location. R, rightward; L, leftward; AS, antisense. PSI, PFA sensitivity index. TATW (W = A or T) positioned -37 to -29 upstream of TSS.

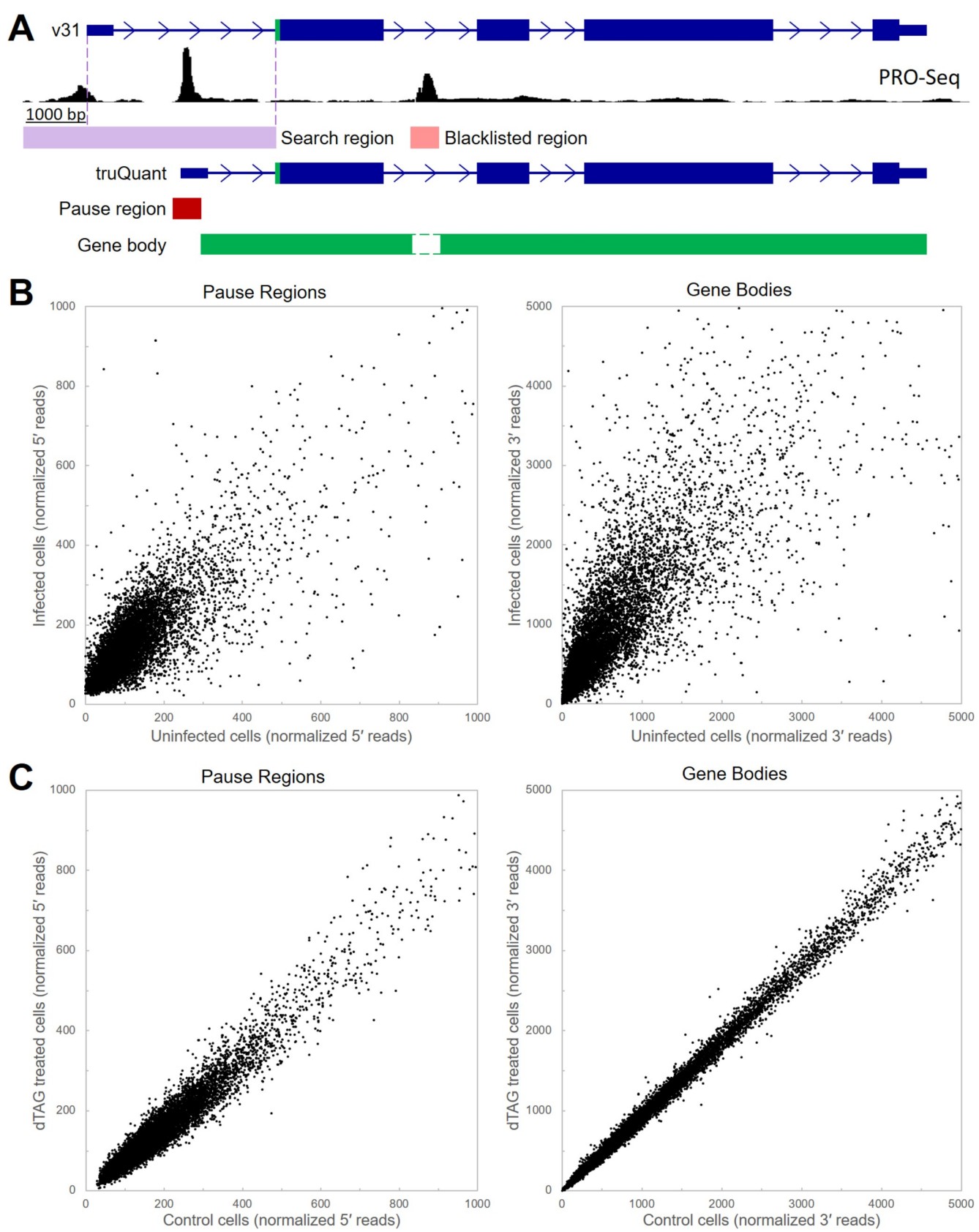

**Fig 6. Differential expression analysis of HCMV TB40/E infection and IE2F depletion using truQuant. (A)** Schematic of truQuant method modifying GENCODE v31 basic protein coding annotations. TSRs were identified in a search region extending 1,000 bp upstream of the 5′ end of v31 to the most upstream start codon in order to create truQuant annotations with 5′ ends as the maxTSS within the maxTSR. Pause regions were defined as a ±75 from the annotated 5′ end of the gene and quantified by summing the 5′ ends of reads in the region. Non maxTSRs were expanded and blacklisted for 3′ ends for gene body analysis by summing the 3′ ends of reads from the end of the pause region to the 3′ end of the gene. Correlation plots for **(B)** the effect of infection and **(C)** the effect of IE2F depletion.

The IE2-activated TSR for the UL83 promoter, which is considered an early-late promoter, has a PSI 28.8. The PSI scores are lower for 24 other IE2-activated TSRs. Approximately 76% of all IE2-activated TSRs lack a TATT or TATA positioned at -37 to -30 upstream of the TSS. This includes the UL146 TSR that is activated the most by IE2 (40.5-fold) and has a PSI 37.4. Five of the 42 IE2-activated TSRs have an upstream TATA, with all but 1 of these 5 TSRs having relatively low PSIs. Five of the 42 IE2-activated TSRs have an upstream TATT with PSIs scoring in the lateness range of IE2-unaffected TSRs having an upstream TATT. Among this group of 5 is a TSR at 382 bp upstream of the UL147A ORF that has a PSI 318.1 that is comparable to the PSI for the TSR of the prototypical TATT-containing UL99 (pp28) late promoter. We were struck by the finding that a group of TSRs with an upstream TATT that are unaffected by IE2 depletion have PSIs in the range of viral early-late promoters. Examples of this group include the TSRs for the early-late UL22A, UL84, and UL98 genes [48–50]. The UL22A, UL84, and UL98 promoter TSRs have the upstream sequences 5'-TATTTAACGGCA-3' at -33, 5'-CATATTTAAAGG-3' at -34, and 5'-TATTTATATAAC-3' at -34, respectively. The tandem linkage of TATT and TATA sequences upstream of the putative TSS for the UL98 alkaline nuclease promoter might be one way of imparting the early-late, lower PSI trait to transcription control.

Having established the effects of IE2 on viral transcription, we turned to an examination of potential effects on transcription of the host genome in late infection. To obtain a quantitative measurement of the levels of promoter-proximal pausing as well as productive elongation we developed truQuant, a new tool to analyze PRO-Seq data (**Fig 6A**). The analysis starts with reannotation of the host genome using PRO-Seq data from the cells being analyzed to accurately identify the most highly utilized TSS for each gene that is expressed. A sum of all reads with 5′ ends falling in the middle of a 151 bp region surrounding the main TSS provides quantification of the amount of pausing. The sum of reads with 3′ ends within the downstream region up to the transcription end site (usually the major Poly(A) addition site) quantifies productive elongation through the gene body. To increase accuracy, regions of transcription initiation (alternative minor start sites and enhancers) within the gene bodies were identified and removed from the quantification of productive elongation. To validate the ability of truQuant to discover changes in transcription, we first analyzed PRO-Seq datasets from HFF before and after a 72 h HCMV TB40/E infection. As expected, large changes in the transcription of hundreds of host genes were found. This is seen as points off the diagonal in correlation plots of reads from both pause regions and gene bodies from 9,942 expressed genes identified in the infected cells (**Fig 6B**). There was a global 1.7-fold increase in the pausing signal and a 2.4-fold increase in productive elongation indicating that HCMV infection causes an overall increase in transcription by Pol II. The larger fold increase in productive elongation over pausing suggests infection leads to an increase in the function of P-TEFb. The relative fold change in transcription of each gene was determined by dividing each individual fold change by the global average. This revealed a 3-fold or greater change over the pause region for 518 genes (183 up and 335 down). Likewise, 842 genes displayed a 3-fold or greater change over the gene body (205 up and 637 down) (**S2 Data**. Many of the genes with the largest fold changes have been previously identified as antiviral response genes. Examination of the read normalized browser tracks provides examples of large increases in both paused and productively elongating Pol II

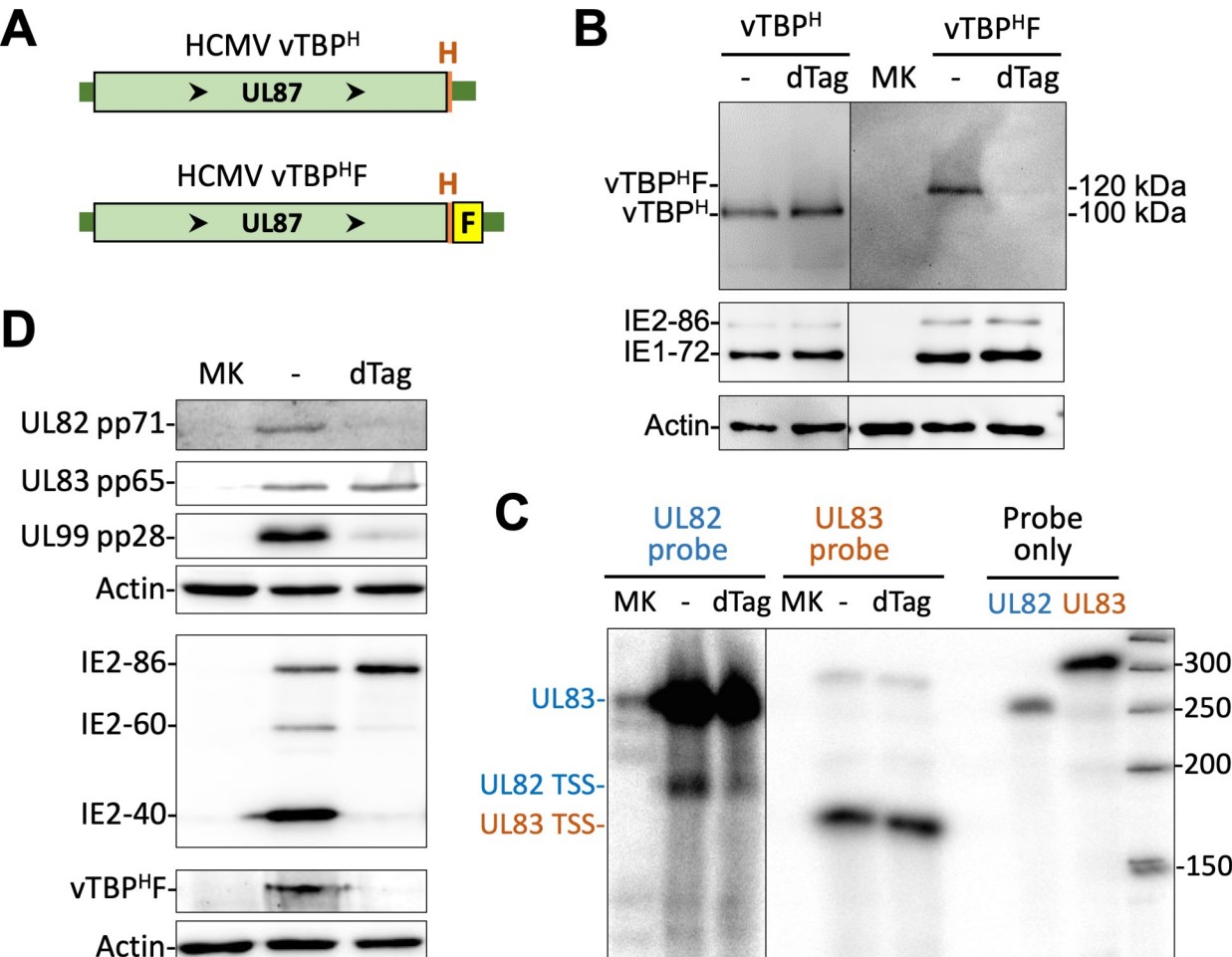

**Fig 7. vTBP depletion does not affect UL83 promoter-dependent expression in late infection. (A)** The carboxy end of vTBP, encoded by UL87, was tagged with the HA epitope alone (vTBP$^H$) or HA plus FKBP12M (vTBP$^H$F) in the HCMV Towne genome. **(B)** HCMV vTBP$^H$- and vTBP$^H$F-infected HFF tested in parallel were analyzed by western blot for the indicated proteins at 72 h pi plus or minus dTAG at 66–72 h pi. Different imaging exposure times were used to view vTBP$^H$- and vTBP$^H$F bands. **(C)** HCMV vTBP$^H$F-infected HFF were analyzed by RPA at 72 h pi plus or minus 6h-dTAG, using riboprobes depicted in Fig 3B, and bands were quantified using a phosphorimager. **(D)** At 72 h pi with HCMV vTBP$^H$F, dTag was added or not for 24 h and then the indicated proteins were analyzed by western blot.

(ISG15, OASL, and C12orf43) and a large decrease (TXNIP) (S4A Fig). Having proven that truQuant is useful in identifying changes in transcription between datasets we then compared the data from spike-in normalized PRO-Seq data for TB40/E IE2F-infected HFF with and without dTag-mediated depletion of IE2F. As evidenced by the correlation plots there were very few changes in transcription (Fig 6C) with none that had a 2-fold change in the both pausing region and the gene body (S2 Data). BIK, EID2B, and CPED1 had some of the largest changes after IE2F depletion, but genome browser tracks demonstrate that the effects were very modest (S4B Fig). We conclude that depletion of IE2F late in infection does not have any substantial effect on transcription of the host genome by Pol II.

## vTBP activates the late UL82 promoter but not the IE2-activated UL83 promoter

In late infection, viral DNA replication begets or strengthens transcription from many IE2-activated promoters yet not all of these promoters have an upstream TATT. To substantiate the

idea that vTBP is not universally required for transcription at all the IE2-activated promoters, we examined the effect of targeted vTBP degradation on RNA expression from the TATT-less UL83 promoter. The FKBP12M tag along with an HA epitope were fused in-frame to the carboxy end of vTBP in the Towne strain genome to create HCMV vTBP$^H$F (**Fig 7A**, **S5 Fig**). Treatment of HCMV vTBP$^H$F with 200 nM dTag for 6 h at 66–72 h pi, greatly decreased the amount of vTBP$^H$F protein, but had no effect on amount of the HA-tagged vTBP expressed from the control HCMV vTBP$^H$ virus (**Fig 7B**). Assessment by RPA of the outcome of the vTBP$^H$F depletion over this timeframe revealed no change in level of UL83 RNA emanating from the UL83 promoter, while the level of UL82 promoter-generated RNA decreased by ~50% (**Fig 7C**). Lengthening the duration of dTag treatment to 24 h was also ineffective at changing the amount of the UL83 pp65 protein produced, but greatly lowered levels of the late UL82 and UL99 gene products, pp71 and pp28, respectively (**Fig 7D**). This latter result accords with the finding that UL82 and UL99 each have a TATT positioned upstream of the TSS. The presence of TATT in the late promoters driving IE2-40 and IE2-60 expression also explains the drop in IE2-40 and IE2-60 protein levels resulting from the 24h-depletion of vTBP$^H$F. This was accompanied by a modest increase in IE2-86 amount, possibly because less IE2-40 and IE2-60 imparts less blockade to MIEP transcription. Remarkably, maintaining IE2-86 abundance while greatly depleting IE2-40 and IE2-60 by targeted vTBP$^H$F degradation had not altered UL83 expression. This implies that IE2-86 alone may be sufficient to drive UL83 promoter activity.

## Discussion

The IE2-40, IE2-60, and IE2-86 isoforms have been previously determined to have overlapping functions using assays having major limitations for predicting whether these proteins promote viral transcription in late infection [9]. All of the isoforms have the protein region that binds to DNA, promotes dimerization, and interacts with various host and viral proteins [51]. This potential complexity provided the rationale for disposing of all IE2 family members in late infection to determine their regulatory role in viral transcription. In this report, we have leveraged recent technological advances in targeted protein degradation and global analysis coupled with multiple validation methods to provide compelling evidence that the IE2 proteins drive transcription from a select set of viral promoters in late infection. IE2 greatly enhances the engagement of promoter proximal Pol II at these promoters. This outcome is possibly the result of a strengthening of preinitiation complex assembly or loading onto the promoters.

Viral early, early-late, and late genes are classified according to temporal pattern of RNA or protein expression in relation to onset and need of viral DNA replication. Our quantitative analyses of nascent RNAs initiating at individual TSSs provides a new perspective on the relationship between change in transcription pattern and conventional kinetic class determination. The studies herein expose the prevalent and often differential use of 2 or more TSSs to initiate transcription of the same gene over the course of infection. As an example, the two TSSs in the promoter region of the UL82 (pp71) gene are separated by 65 nucleotides but function in completely opposite ways. One TSS is active before viral DNA replication at 24 h pi and has an upstream TATAA sequence, whereas the other TSS is active after onset of viral DNA replication at 72 h pi and has an upstream TATTT. The broader implication of these findings is that temporal change in steady-state levels of many mono or polycistronic viral RNAs may be the sum output of 2 or more independent TSSs. This added to the rationale of selecting the method of RPA to quantify change in UL82 TSS output for validating the PRO-Seq findings. The studies also reveal the ambiguity in distinguishing between early and early-late transcription. Individual TSSs rank differently in the ratio of amounts of nascent RNAs they produce

before and after the amplification of viral DNA template. The PSI rankings for IE2-activated TSRs not having an upstream TATT are reflective of this continuum in degree of lateness that ostensibly encompasses the categories of early, early-late, and late kinetic-class promoters. In late infection, at least 10 of 42 IE2-activated TSRs have PSI lateness rankings that are comparable to those for the bulk of the TATT-linked TSS population that is unaffected by IE2. These IE2-activated TSRs are negligibly active in early infection at 24 h pi. Half of the 10 IE2-activated TSRs in this group have an upstream TATT that could possibly be a target of vTBP, whereas the other TSRs in the group do not have an upstream TATT or TATA. The finding that a betaherpesvirus has a subset of TATT-less viral late promoters that are probably not dependent on viral LTFs for transcription aligns with the recent reports that Epstein-Barr virus (EBV), a gammaherpesvirus, also has a class of viral late promoters that do not depend on viral LTF for transcription [52, 53]. The mechanism underlying the activation of the LTF-independent EBV late promoters is unknown, but the viral lytic switch proteins could conceivably be involved.

The viral UL83 and UL146 genes exhibit early-late kinetics based on prior measurements of steady-state RNA levels in relation to inhibition of viral DNA replication and time after infection [45, 54]. Our results show that IE2 strengthens productivity of viral UL83 and UL146 TSRs in late infection. These TSRs lack an upstream TATT or TATA. The PSIs for UL83 and UL146 TSRs score in the lateness range of the TATT-linked TSRs juxtaposed upstream of UL84 and UL22A genes, which are not affected by IE2. The TATT-linked TSR of the UL98 alkaline nuclease gene that is insensitive to IE2 has a PSI score lower than that of 76% of all IE2-activated TSRs. Having a TATA linked in tandem with the TATT of UL98 might enable TSS use before the full complement of viral LTFs are present. We do not yet actually know the extent to which vTBP and other LTFs are involved in the activation of TATT-linked TSSs, irrespective of dependency on IE2. Nevertheless, a considerable number of viral TSSs have an activity profile consistent with early-late gene kinetics and most are unaffected by IE2 depletion in late infection.

Three-quarters of the IE2-activated TSRs do not have a TATT or TATA positioned at -37 to -30 upstream of the TSS. Widening the window of consideration from -40 to -26 upstream of the TSS increased by 1 the number of TSRs having an upstream TATA. None of the 16 TSRs activated 5-fold or greater by IE2 have an upstream TATT or TATA. We therefore conclude that neither TATT nor TATA is universally required for the IE2-dependent initiation of transcription in late infection. Because the methodologic approach applied in these studies does not eliminate 100% of existing IE2 proteins over a 6-h timeframe, our list of viral promoters regulated by IE2 in late infection may be incomplete. The quantitative effects of IE2 are also likely to be greater than what is shown due to some remaining IE2.

Exactly how IE2 drives transcription initiation at select viral promoters is unclear. The selection may result from IE2 binding directly to sites in the viral genome or to host proteins (or another viral protein) that bind to specific sites in the viral promoter. A *crs*-like sequence to which IE2 is known to bind is not located in or nearby many of the IE2-activated promoters, but non-*crs*-like elements can also be bound by IE2 as has been described for HCMV early promoters [55, 56]. Alternatively, IE2 might bring about transcription from a set of weakly active viral promoters as a consequence of the onset of viral DNA replication that rendered the genome more accessible to host GTF that interact with IE2. For the HSV alphaherpesvirus, a single round of viral genome replication permanently makes the viral genome more accessible to host preinitiation complex components (TBP, TAF1, and Pol II) and enables the immediate early ICP4 protein to increase Pol II occupancy at viral late promoters [22]. The individual contribution of the different HCMV IE2 isoforms in the transcription activation remains to be determined. For the 5 IE2-activated TSRs that have TATTs positioned upstream, the extended

sequences of these TATTs deviate somewhat from the TATTWAA core sequence that is the favored binding site for the vTBP protein of the human gammaherpesvirus Kaposi's sarcoma-associated herpesvirus (KSHV) [57]. The HCMV vTBP-dependent UL82 promoter has a TATTTGC sequence suitably positioned upstream of the TSS that also deviates from the KSHV TATTWAA core sequence. This suggests that beta- and gammaherpesviruses might differ in the promoter sequences preferred by their respective vTBPs.

Pre-empting viral DNA synthesis by applying the PFA inhibitor at the beginning of infection prevents viral late promoter transcription. Inhibiting viral DNA replication for 6 h in late infection by applying PFA for the same timeframe selected for IE2 depletion did not appreciably alter any viral transcription. This latter finding indicates that the change in viral transcription caused by depleting IE2 for 6 h in late infection is unlikely the result of inhibiting viral DNA replication-fork progression, which might occur from disrupting IE2's putative interaction with the viral DNA polymerase processivity factor (UL44 protein) [58]. Based also on this finding, we surmise that at least a fraction of the already produced viral DNA genome replicates is enduringly rendered permissive to Pol II transcription (re)-initiation at viral late promoters.

Repressive histones are unlikely to occupy HCMV DNA templates that yield pervasive transcription. Herpesviruses have evolved a variety of strategies to escape the loading of histones onto the viral genomes that undergo amplification. Viral immediate-early proteins may participate in this process. For example, the HSV ICP4 protein changes core histone dynamics in ways that may mobilize histones away from the viral genome [59] and HCMV IE2 counteracts the repressive effect of histone H1 [14]. In contrast, the host genome remains highly associated with histones organized in nucleosomes. This level of chromatization may account for our observation that HCMV IE2 (and possibly LTFs [33]) does not activate promoters in the host genome during late infection. Our results are consistent with the hypothesis that at least a fraction of replicated HCMV genomes are rid of repressive chromatin to allow IE2 and LTFs to activate the late infection viral promoters. Regardless of the role that chromatin plays, we deduce that HCMV IE2 and LTFs apply two different strategies to carry out their directives—IE2 uses an alphaherpesvirus-like approach and LTFs use a gammaherpesvirus-like approach.

## Materials and methods

### Cells, viruses, and plasmids

HFF were isolated from de-identified discarded human foreskins and propagated in Minimum Essential Medium (Gibco, 11095080) supplemented with heat-inactivated 5% fetal bovine serum (Gibco, 26140079) and 1% penicillin-streptomycin (Gibco, 15140122). HFF were studied at passage number ≤6. HCMV Towne BAC[7] and TB40/E BAC4 [60] (kind gift from E. Murphy) were used in this study. These BAC viruses were used to construct HCMV recombinant viruses, Towne IE2F, TB40/E IE2F, Towne vTBP$^H$ and Towne vTBP$^H$F. All HCMV recombinant viruses were constructed using the galK selection method described by Warming et al [61]. The sequences of the primers used for constructing recombinant viruses and plasmids are listed in supplemental **S2 Table**. Two steps of homologous recombination were used to construct the recombinant HCMV BAC viruses. Briefly, the bacterial galK gene was placed at the carboxy-termini of IE2 and UL87 (vTBP) ORFs. PAGE-purified oligos were used in PCR to generate products containing galK and the indicated flanking viral sequences. The PCR products were electroporated into SW105 *E. coli* containing Towne WT BAC or TB40/E WT BAC4, and then grown under chloramphenicol and galactose selection. In the next step, the galK gene was replaced by the FK506 binding domain protein 12 ORF containing the F36V amino acid substitution (FKBP12M) fused in-frame with the carboxy terminus of the

IE2 or vTBP ORF. The hemagglutinin (HA) epitope (YPYDVPDYA-L-protein) was fused to the carboxy terminus of the vTBP ORF either alone (vTBP$^H$) or together with FKBPM12M (vTBP$^H$F). The tag was added in a manner that should not disrupt UL88 reading frame. The pDONR221_FKBP12m plasmid was the source of the FKBP12m. Chloramphenicol and 2-deoxy-galactose counterselection was used to recover the recombinant. All BAC viruses were stored in DH10B *E. coli*. HCMV BACs were transfected into HFF using the Amaxa Nucleofector II at program setting U23 and the Amaxa Neonatal Human Dermal Fibroblasts (NHDF) kit (Lonza, VPD-1001). In all cases, DNA sequencing through the modified IE2 and vTBP regions confirmed the intended changes, and the restriction fragment length polymorphism analysis substantiated the integrity of the recombinant HCMV genomes. Because of pervasive transcription of the viral genome in late infection, PRO-Seq produced a high depth of read coverage across the viral genome that enabled the determination that spurious deletions or insertions had not occurred in the HCMV BAC genomes. All of the HCMV BAC viruses express green fluorescent protein (GFP) from a simian virus 40 (SV40) early promoter-GFP gene located in the BAC DNA segment.

Plasmids pGEM4-UL82p and pGEM4-UL83p contain 260-bp UL82 and 320-bp UL83 promoter segments, respectively, that straddle the TSS of the corresponding promoter. The promoter segments HCMV Towne genome were PCR-amplified using primer sets listed in S2 Table. The ends of the PCR products were trimmed with *Nsi*I and *Eco*RI prior to ligation of the promoter segment into the *Pst*I and *Eco*RI sites of pGEM-4Z (Promega).

All viruses were isolated from infected cell supernatant that was passed through a 0.45 μm filter prior to pelleting virus through a 20% sorbitol cushion [62]. Once the nucleofected HCMV TB40/E BAC DNA produced ample amount of virus in HFF, the virus was passed onto ARPE19 cells to make viral stock aliquots that were used to inoculate HFF to make a larger viral stock for experiments. Minimizing the number of TB40/E replication cycles in HFF and growing the virus in ARPE19 cells helps preserve the phenotype-genotype of this virus. All viruses studied in parallel had been propagated and tittered in parallel. Viral stocks were serially diluted to attain an MOI of 0.5–0.8 infectious units per HFF cell as determined by immunofluorescence assay of IE1/IE2 protein-positive cells at 24 h pi, using murine monoclonal antibody MAB810 (EMD Millipore, 1:1000 dilution) followed by a secondary goat anti-mouse IgG (H+L) antibody conjugated to Alexa Fluor 555 (Thermo Fisher Scientific, A-21422). The results informed the calculation of the number of infectious units per mL of viral inoculum [63, 64].

## Infections and treatments

All experiments were carried out using confluent HFF infected at MOI of 1–3 and maintained in Minimum Essential Medium containing 5% fetal bovine serum. For experiments comparing different viruses, the titer of the viral inoculum was adjusted so that the amount of infectious virus that was applied to the cells was the same for the different viruses. Only viral stocks differing in infectious titer by ≤4-fold were used in comparative analyses. HFF derived from 3 different donors were used to generate four separate PRO-Seq datasets. The degradation tag compound, dTag-7, was kindly provided by Bradner, Winter, and Gray [43] and dissolved in DMSO. For precision nuclear run-on and sequencing [40] (PRO-Seq) experiments involving dTag treatment, a set of 3 HFF infections were carried out in parallel to perform PRO-Seq, western blot, and RT-qPCR assays. PRO-Seq was performed after there was evidence from western blot and RT-qPCR results that the targeted degradation system was working as intended. For each PRO-Seq test condition, the culture medium was refreshed 24 h prior to infection of the HFF monolayer in the T-150 cm$^2$ flask. On the day of infection, all but 6 mL of

the conditioned medium was removed from the cells into which virus was inoculated in an amount for MOI of 3. The culture medium that had been removed was re-applied after the 60 min viral adsorption period. At the indicated times in late infection, all but 6 mL of the conditioned culture medium was removed from the infected cells. To 6 mL of conditioned medium that had been removed, 6 μL of 400 nM dTag-7 in DMSO or DMSO alone was added. This 6 mL of medium was then mixed with the 6 mL in the T-150 cm$^2$ flask to produce a final concentration of 200 nM dTag-7. For PRO-Seq analyses involving treatment with flavopiridol (Flavo) versus DMSO vehicle during the last 1 h of infection, 6 mL of conditioned medium was removed from each of the T-150 cm$^2$ flasks and spiked with 6 μl of 2 mM Flavo (NIH AIDS Reagent Program 9925z) in DMSO or DMSO alone. The medium was then returned immediately to the flask for a total of 12 mL of medium having a final 1 μM Flavo in the Flavo-treated infections. Phosphonoformic acid (PFA) (Sigma-Aldrich) was added to the medium at the beginning of the 72 h TB40/E infection in a final concentration of 400 μg/mL. Prior to using the sample for PRO-Seq, aliquots of the isolated infected nuclei from PFA-treated and -untreated HFF were subjected to qPCR to determine that PFA lowered the number of viral DNA copies by 100-fold at 72 h pi compared to the untreated infected nuclei. In TB40/E experiment 1, PFA was added for 6 h in late infection in the same way the dTag was added to permit comparison of PFA and dTag treatment results. The final PFA concentration was 400 μg/mL in a total of 12 mL of medium in a T-150 cm$^2$ flask.

## RT-qPCR analysis

TRIzol Reagent was used to isolate whole-cell RNA according to the manufacturer's instructions (Invitrogen, 15596026). Reverse transcription (RT) and quantitative real-time PCR (qPCR) were performed using methods described previously [63] and PCR primers listed in **S2 Table**. The IE1, IE2, MIE, and UL128 primer sets span the introns for their respective genes. All RT-qPCR analyses were performed in triplicate (triplicate infections, triplicate treatment conditions) in 12-well plates and results normalized to host GAPDH RNA. Standard curve method was applied in all RT-qPCR analyses. The 7500 Fast Real-Time PCR System (Thermo Fisher) was used throughout using the same PCR parameters: 95˚C for 10 min, followed by 40 cycles at 95˚C for 15 s, then 60˚C for 60 s.

## RNAse protection assay

The RNase protection assay (RPA) was performed on whole-cell RNA, as described previously [64]. pGEM4-UL82p and pGEM4-UL83p were linearized with *Eco*RI and gel purified. Antisense UL82 and UL83 $^{32}$P-labeled riboprobes were generated from the linear pGEM4-UL82p and pGEM4-UL83p templates, respectively, using SP6 polymerase (Agilent, 600151–1), alpha-$^{32}$P uridine 5'-triphosphate (3000 Ci/mmol, Perkin Elmer, BLU007H250UC), and RNase OUT Ribonuclease Inhibitor (Invitrogen). The UL82 probe is predicted to protect a 180-nt RNA product arising from the UL82 promoter TSS and a ~260-nt UL83 RNA product running through the UL82 gene body [46]. The UL83 probe is predicted to protect a 160-nt RNA product arising from the UL83 promoter TSS. The hybridization of riboprobe(s) to the RNA sample (20 μg) was performed overnight at 52˚C, and the resultant hybrid sample was digested with 1000 U of RNase T1 (Thermo Scientific, 1000 units/μL, EN0541) at 37˚C for 1 h. Protected products were analyzed on 6% polyacrylamide-urea gels. The marker was made from 50 bp DNA ladder (Invitrogen, 10416014). 1 ug of this DNA was dephosphorylated in a 20 μL volume using Antarctic phosphatase (New England BioLabs, M0289S) for 30 min at 37˚C, according the manufacturer's directions. Heat inactivation was carried out for 5 min at 80˚C. The DNA was later denatured at 90˚C and plunged on ice. A 5 μL volume of this DNA

was used in a 10 μL reaction containing 1 μL T4 DNA polynucleotide kinase (PNK) (New England BioLabs, M0201S), 1 μL 10x PNK buffer, and 3 μL of gamma-[32]P Easy Tides adenosine 5'-triphosphate (6000 Ci/mmol, Perkin Elmer, BLU502Z250UC) incubated for 10 min at 37°C. The reaction was stopped by adding 1 μL 0.5M EDTA and heating at 65°C for 20 min. The volume was increased to 100 μL with H2O and subjected to phenol-chloroform extraction. The aqueous layer was removed and 50 μL 7.5 M ammonia acetate and 200 μL of 95% ethanol was added to precipitate the DNA fragments at 70°C for ≥30 min. After centrifugation, the pellet was washed with 95% ethanol, dried, and resuspended and stored in 50 μL of RNA loading buffer (90% formamide, 10 mM EDTA, and 0.2% bromophenol blue).

## Western blot

Western blot was performed on sonicated whole-cell extracts in lysis buffer containing phosphatase and protease inhibitors, as previously described [65]. The gel-loading buffer contained 2% SDS and 100 mM beta-mercaptoethanol. The proteins were fractionated on freshly made 8–10% SDS-PAGE Tris-glycine gels prior to transfer to Amersham Protran 0.45-μm nitrocellulose membranes (GE Healthcare Life Sciences 10600002). HCMV IE2-p60, IE2-p40 and IE2-p86 were detected by monoclonal mouse antibody MAB8140 (Millipore sigma, 1:1,000 dilution). pp28 was detected by mouse monoclonal CMV p28 UL99 antibody (Fitzgerald, 1:500 dilution). pp71 were detected by polyclonal goat CMV pp71 antibody (Santa Cruz Biotechnology, vC-20, 1:200 dilution). pp65 was detected by mouse monoclonal CMV pp65 antibody (Fitzgerald, 1:1,000 dilution). HA (vTBP[H] and vTBP[H]F) was detected by monoclonal mouse anti-HA.11 epitope tag antibody (BioLegend, 1:1000 dilution). IE1-p72 and IE2-p86 were detected by murine monoclonal antibody MAB810 (EMD Millipore, 1:1,000 dilution). Host actin was detected with polyclonal rabbit anti-actin antibody (Sigma-Aldrich, A2066, 1:4,000 dilution). Primary antibodies were detected with peroxidase AffiniPure F(ab')$_2$ fragment rabbit anti-goat IgG (H+L) (Jackson ImmunoResearch, 305-036-045, 1:40,000 dilution), peroxidase AffiniPure F(ab')$_2$ fragment goat anti-mouse IgG (Jackson ImmunoResearch, 115-036-006, 1:40,000 dilution), rabbit anti-mouse IgG (whole molecule)–peroxidase antibody (Sigma-Aldrich, A9044, 1:40000 dilution), or goat anti-rabbit IgG (whole molecule)–peroxidase antibody (Sigma-Aldrich, , A0545, 1:40,000 dilution). Blots were treated with SuperSignal West Femto maximum-sensitivity substrate (Thermo Scientific SJ257615). Blots were imaged and analyzed with the UVP ChemStudio Imaging System (Analytik Jena).

## Viral DNA replication assay

HFF were infected in triplicate with Towne IE2 or IE2F at the indicated MOI. The cells were washed thrice after adsorption of the viruses and harvested at he indicated times post-infection. The cells were washed with 1x PBS just prior to harvesting in PCR lysis buffer (10 mM Tris-HCl [pH 8.0], 1 mM EDTA, 0.001% Triton X-100, 0.0001% SDS) containing 20 μg/mL proteinase K. The lysate was incubated 55° C for 100 min and the proteinase K was then heat-inactivated at 95°C for 20 min, as previously described [64]. Viral DNA was quantified by real-time PCR using primers targeting IE1 exon 4 using primers listed in **S2 Table** and the PCR parameters described for RT-qPCR.

## PRO-Seq library preparation

Each PRO-Seq library was derived from a single T-150 flask of HFF infected at MOI of 3. PRO-Seq libraries were prepared as described in Parida et. al. [33] with some modifications. Infected and drug-treated human foreskin fibroblasts were lysed and nuclei harvested and stored as previously described [44]. Nuclei were gently pelleted and storage buffer was

removed. Nuclei were then resuspended in 40 μL of a buffer containing 20 mM HEPES (pH 7.8), 5 mM magnesium chloride, 100 mM potassium chloride, 5 mM dithiothreitol (DTT), and 0.6 U/μL SUPERase-In (Invitrogen AM2696). Spike-in controls were utilized in this study and spike-in nuclei were introduced at one of two different stages of the library preparation process. For the TB40/E PRO-Seq datasets, approximately 100,000 moth Sf21 cells (of *Spodoptera frugiperda* origin) were added to the cell lysates prior to passing the nuclei through a sucrose cushion and resuspension in storage buffer. For the 24 h pi, 72 h pi, and 72 h pi PFA ± Flavo TB40/E PRO-Seq datasets, approximately 100,000 Sf21 nuclei, which were isolated in a manner identical to the way in which HFF nuclei were isolated, were combined with the HFF nuclei prior to being spun down and resuspended. Measurements of Sf21 cell and nuclei concentration were performed using a Countess (Thermo Scientific). The details of the nuclear walk-on method have been published previously [33]. Briefly, resuspended nuclei were warmed to 37˚C and then combined with 20 μL of a 3X nuclear run-on mix containing 20 mM HEPES, pH 7.8, 5 mM magnesium chloride, 100 mM potassium chloride, 5 mM DTT, 1.5% Sarkosyl, and 60 uM of biotinylated ATP, UTP, GTP, and CTP (Perkin Elmer NEL544, NEL543, NEL545, and NEL542, respectively). Samples were quickly mixed after the addition of nuclear run-on mix and allowed to incubate at 37˚C for 10 minutes. Reactions were quenched with the consecutive addition of 40 μL of 50 mM ethylenediaminetetraacetic acid (EDTA) and 300 μL of Trizol LS (Ambion 10296028). Trizol RNA extraction was carried out according to the manufacturer's protocol. Subsequent steps in library preparation, from RNA base hydrolysis, streptavidin-affinity selection of biotinylated RNA, 3' adapter ligation, 5' adapter ligation, through reverse transcription were carried out as previously described [33]. Notably, while preparing the Towne PRO-Seq libraries for this work, we discovered during our analysis of other HCMV PRO-Seq libraries that unique molecular identifiers (UMIs) of 4 nucleotides in length positioned at the ends of the RNA adapters provided insufficient redundancy to capture all nascent RNAs with the same 5' and 3' end, particularly those associated with the HCMV RNA 4.9 promoter-proximal pause region. All other PRO-Seq libraries prepared for this study utilized RNA adapters that each contain 8-nucleotide UMIs. Reverse transcription was carried out in a final volume of 23 μL to produce library cDNA. A 2 μL volume of each library was serially diluted and subjected to 24 cycles of PCR amplification using 1.5 μM of RP1, 1.5 μM Illumina barcoded index primer RPI-6 (see below), and 12.5 μL KAPA HiFi HotStart ReadyMix (KAPA Biosystems KK2601) in total volume of 25 μL and the following PCR parameters: 98˚C 45 s; 24X 98˚C– 15 s, 60˚C– 30 s, 72˚C– 30 s; 72˚C– 1 min; and 4˚C–hold. Illumina barcoded index primers were synthesized and HPLC-purified by Integrated DNA Technologies. PCR products were examined by electrophoresis on a TAE 6% acrylamide gel. The number of PCR cycles needed to optimally produce PCR products of the appropriate size distribution was determined for each library. This number ranged from 11– 13 cycles for Towne, 13–14 cycles for TB40/E experiment 1, 14–16 cycles for TB40/E experiment 2, and 15–17 cycles for TB40/E experiment 3. The optimal PCR cycle number was applied in PCR amplification of the 21 μL volume of remaining cDNA library. The PCR reaction included 1.5 μM RP1, 1.5 μM of an Illumina barcoded index primer RPI 1–11 (the barcode differs between the libraries), and 25 μL KAPA HiFi HotStart ReadyMix in a total volume of 50 μL. The PCR products were subsequently purified with a Qiagen MinElute kit in 30 μL. Concentration of the purified cDNA library was determined by the Qubit high-sensitivity dsDNA assay, and a 2 μL sample of each cDNA library was diluted to 2.5 ng/μL and cDNA size distribution analyzed using an Agilent Bioanalyzer 2100. Libraries were pooled in equimolar quantities and size-selected on a Sage Science Blue Pippen instrument using a 2% agarose gel cassette (BDF2010) with Marker V1 internal standards. For the Towne PRO-Seq datasets that utilized 4N UMIs on RNA adapters, library fragments between 135–600 bp in length were

selected. For all other libraries, which were prepared using 8N UMI RNA adapters, fragments between 145–600 bp in length were selected. Pooled library recovered from the size-selection was purified on a Qiagen MinElute column, DNA concentration was measured as previously described using a Qubit, and a sample of the library was diluted to 3 ng/μL for analysis on the Bioanalyzer. Upon confirmation that the pooled library was accurately size-selected, the PRO-Seq library was sequenced on an Illumina HiSeq 4000 with 150 bp paired-end reads. All sequencing was performed at the University of Iowa Genomics Division. The Towne, TB40/E IE2F experiment 1, TB40/E IE2F experiment 2, and TB40/E experiment 3 (24 h pi, 72 h pi, and 72 h pi plus PFA ± Flavo) PRO-Seq libraries were sequenced on separate lanes and at different times.

## Adapter and Illumina index primer sequences

VRA3-4N: /5Phos/rNrNrNrNrGrArUrCrGrUrCrGrGrArCrUrGrUrArGrArArCrUrCrUrGrArArC/3InvdT/

VRA5-4N: rCrCrUrUrGrGrCrArCrCrCrGrArGrArArUrUrCrCrArNrNrNrN

VRA3-8N: /5Phos/rNrNrNrNrNrNrNrNrGrArUrCrGrUrCrGrGrArCrUrGrUrArGrArArCrUrCrUrGrArArC/3InvdT/

VRA5-8N: rCrCrUrUrGrGrCrArCrCrCrGrArGrArArUrUrCrCrArNrNrNrNrNrNrNrN

RP1: AATGATACGGCGACCACCGAGATCTACACGTTCAGAGTTCTACAGTCCGA

RPI-1 [ATCACG]: CAAGCAGAAGACGGCATACGAGATCGTGATGTGACTGGAGTTCCTTGGCACCCGAGAATTCCA

RPI-2 [CGATGT]: CAAGCAGAAGACGGCATACGAGATACATCGGTGACTGGAGTTCCTTGGCACCCGAGAATTCCA

RPI-3 [TTAGGC]: CAAGCAGAAGACGGCATACGAGATGCCTAAGTGACTGGAGTTCCTTGGCACCCGAGAATTCCA

RPI-4 [TGACCA]: CAAGCAGAAGACGGCATACGAGATTGGTCAGTGACTGGAGTTCCTTGGCACCCGAGAATTCCA

RPI-5 [ACAGTG]: CAAGCAGAAGACGGCATACGAGATCACTGTGTGACTGGAGTTCCTTGGCACCCGAGAATTCCA

RPI-6 [GCCAAT]: CAAGCAGAAGACGGCATACGAGATATTGGCGTGACTGGAGTTCCTTGGCACCCGAGAATTCCA

RPI-7 [CAGATC]: CAAGCAGAAGACGGCATACGAGATGATCTGGTGACTGGAGTTCCTTGGCACCCGAGAATTCCA

RPI-8 [ACTTGA]: CAAGCAGAAGACGGCATACGAGATTCAAGTGTGACTGGAGTTCCTTGGCACCCGAGAATTCCA

RPI-9 [GATCAG]: CAAGCAGAAGACGGCATACGAGATCTGATCGTGACTGGAGTTCCTTGGCACCCGAGAATTCCA

RPI-10 [TAGCTT]: CAAGCAGAAGACGGCATACGAGATAAGCTAGTGACTGGAGTTCCTTGGCACCCGAGAATTCCA

RPI-11 [GGCTAC]: CAAGCAGAAGACGGCATACGAGATGTAGCCGTGACTGGAGTTCCTTGGCACCCGAGAATTCCA

## Bioinformatics

Upon retrieval of demultiplexed sequencing data, fastq files were trimmed of adapters using trim_galore \—paired—small_rna—dont_gzip—quality 0—length 26/34 (trim_galore 0.4.4 https://github.com/FelixKrueger/TrimGalore). Note that the—length parameter was set to 26 for the Towne IE2F PRO-Seq library that was prepared with 4N UMI RNA adapters and 34 for the remaining libraries that used 8N UMI RNA adapters. This change results in the

minimum processed read length being 18 bp regardless of the length of the UMI. For alignment of reads, two different concatenated genomes were utilized. For Towne datasets, an HCMV Towne genome sequence (GenBank accession number FJ616285.1) [66] concatenated to the human genome sequence (UCSC assembly hg38) was utilized. For TB40/E datasets, an HCMV TB40/E genome sequence (GenBank accession number KF297339.1) [67] concatenated to Spodoptera genome (WGS number JQCY02) and human genome (UCSC assembly hg38) sequences were utilized. The concatenated genome fasta files were used to construct a bowtie index using the bowtie-build -f function. Datasets were aligned using bowtie \ —trim5 4/8—trim3 4/8—minins 26/34—maxins 600—fr—best—allow-contain—sam—fullref —threads 80—chunkmbs 5000. The—trim5,—trim3 and–minins parameters were decided by the length of the UMIs on the RNA adapters. Output .sam files were then deduplicated using dedup -n 4/8 -f -p where -n specifies the length of the UMI on each RNA adapter (https://github.com/P-TEFb/dedup). Mapped, deduplicated reads in output .bed files were then parsed out based on the genome to which they aligned and finally converted to bigwigs using the bedtools [68]. The Towne PRO-Seq datasets were normalized to total deduplicated reads mapping to the viral genome (control and Flavo groups were normalized separately). For datasets that incorporated spike-in controls, data were normalized as described in Ball et. al. [69]. Refer to **S1 Table** for details about the calculations performed to normalize the data. Normalization factors were applied via multiplication by genome coverage values in bedGraph files. PRO-Seq reads aligning to annotated HCMV Towne (FJ616285.1) [66] and TB40/E (KF297339.1) [67] genomes were viewed in the UCSC Genome Browser [70] and assembled in Towne and TB40/E track hubs [71], respectively. Both hubs have the open reading frame annotations identified by ribosomal profiling data generated human fibroblasts infected with the HCMV Merlin strain [47]. The additional ORFs identified by this method were located in Towne and TB40/E genomes using Snapgene and represented in the hub as a bigbed file. The TB40/E hub also contains HCMV Merlin mRNA-Seq data at 72 h pi (GSE41605). The two replicates of the Merlin mRNA-Seq datasets (GSM1020246 and GSM1020249) were aligned to the TB40/E genome. The SRA files for these datasets were downloaded using fastq-dump Afterwards, and trim galore was used in succession to remove–-adapter CTGTAGCCACCATC and—adapter AAAAAAAAAA. The trimmed fastq files from the two replicates were concatenated. Afterwards, a hisat2 index for the TB40/E genome was constructed using hisat2-build -f. The trimmed, concatenated fastq file was then mapped to TB40/E using hisat 2 \ -q–threads 80. The output .sam file was converted to a bigwig for display on the TB40/E hub.

## Quantitative analysis of the effect of depletion of IE2F and inhibition of HCMV DNA replication on HCMV transcription

Transcription start regions (TSRs) were identified in PRO-Seq datasets using tsrFinder [33] with predefined parameters applied to the Experiment 1 72 h pi TB40/E IE2F Flavo No dTag dataset. TSRs with >200 total 5′ reads within a 20 bp region were selected from the tsrFinder result. Use of the Flavo dataset more accurately represents TSRs. The number of 5′-end reads at every TSR was determined for 2 different pairs of normalized datasets (Experiment 1 72h IE2F Flavo dTag versus Experiment 1 72h IE2F Flavo No dTag and Experiment 3 72 h No PFA Flavo versus Experiment 3 72 h PFA Flavo). The same steps were followed in the analysis of Towne IE2F datasets comparing Towne IE2F dTag Flavo versus Towne IE2F No dTag Flavo. Scatter plots of the effect of IE2F depletion versus TSR strength were generated for both TB40/E IE2F and Towne IE2F viruses. A PFA sensitivity index (PSI) for TB40/E No PFA Flavo versus PFA Flavo datasets was computed for each TSR as the No PFA/PFA ratio of reads. TSRs with zero counts in the 72 h Flavo PFA dataset were changed to 1 to avoid division by zero

errors. The PSIs range from 0.83 to ≈ 800. The effect of IE2 depletion versus the PSI was plotted for each TSR. TSRs having a TATT sequence with the initial T residing in a region -37 to -30 upstream of the max TSS were identified using bedtools getfasta [68] with the strand-specific flag and a custom python script. The results are provided in the Supplemental Dataset file 1.

## TruQuant differential expression analysis

To create nucleotide resolution annotations for HFF cells infected with the FKPP tagged IE2 TB40-E virus, we first downloaded the GENCODE v31 annotation and extracted the 5′ and 3′ ends of each gene along with the position of the most upstream start codon. The IE2 No dTAG Exp1 dataset was blacklisted using a padded comprehensive RNA list and ran tsrFinder [33]. A program, truQuant was designed to take the modified annotation file, the tsrFinder file, and the blacklisted PRO-Seq data file and create data driven annotations (https://github.com/meierjl/truQuant). Briefly, truQuant works by defining search regions from 1,000 bp upstream of the 5′ end of the gene to the most upstream AUG defined in the modified GENCODE v31 file. Then, it finds all TSRs ([33]) within the search region and reannotates the 5′ end of the gene using the maxTSS from the maxTSR. Pause regions are subsequently defined as a 151 bp window centering on the 5′ end and gene bodies are defined as the end of the pause region to the transcription end site. The effects of infection on pause regions were quantified by summing the 5′ ends of reads in the pause region on the IE2 No dTAG Exp1 file (infected) and an uninfected HFF obtained from Parida, M., et. al (GSE113394) [33]. Gene bodies for both datasets were quantified by summing the 3′ ends in the gene body, excluding reads inside 151 bp regions around TSRs in the gene body that contain at least 30% of the reads of the maxTSR. Computed values were then read normalized according to [69]. The effects of IE2F depletion were analyzed with truQuant using the same annotations, except the resulting values were normalized to the spiked in moth reads. The results are provided in the Supplemental Dataset file 2. To determine the effects of HCMV infection or depletion of IE2F, plots were created to correlate reads counts over the pause region or gene bodies. For analysis of the effect of IE2F depletion, changes in each gene were normalized to the global change to correct for small errors in spike-in control normalizations applied to the raw data.

## Supporting information

**S1 Fig. HCMV IE2F construction. (A)** Map of the HCMV MIE gene locus in relation to *Eco*RI sites and the IE2 proteins expressed from this locus in late infection. The HCMV IE2F BAC construct has FKBP12$^{F36V}$ (F) fused in-frame to the carboxyl ends of IE2 family members, whereas HCMV IE2 BAC has wildtype IE2 proteins. **(B)** HCMV Towne and TB40/E IE2 and IE2F BAC genomes digested with *Eco*RI and fractionated on 0.6% agarose gels. The GalK BACs were derived from IE2 BACs and serve as intermediates for creating the IE2F BACs. Two TB40/E IE2F BACs were derived from separate recombination procedures. The *Eco*RI fragments for IE2 and IE2F are ~10.1 kbp and ~10.4 kbp, respectively. Asterisks mark positions of these *Eco*RI fragments.
(TIF)

**S2 Fig. Comparison of HCMV IE2 and IE2F DNA replication and effect of dTag. (A)** At 72 h pi in HFF, the amounts of genomes for both HCMV Towne IE2 and IE2F viruses (MOI of 3) increased approximately 31-fold, compared to amounts of viral genomes exposed to the PFA inhibitor (200 μM) of viral DNA synthesis throughout the infection. HCMV DNA in triplicate infections was quantified by real-time PCR and normalized to host GAPDH. **(B)** HFF infected

with HCMV Towne IE2F at MOI of 3 were treated with vehicle control (CTRL) or 200 nM dTag at 90–96 h pi. Western blot performed at 96 h pi and band intensities of IE2F-86, IE2F-60, and IE2F-40 determined for dTag relative to CTRL. Mock, MK. **(C)** Effect of 6-h dTag (200 nM) versus CTRL treatment at 90–96 h pi on production of HCMV DNA and cell-associated infectious progeny (MOI 0.5) at 96 h pi. Results of 3 biological replicates. HCMV DNA quantified as in panel A. PFU determined by viral plaque assay on HFF.
(TIF)

**S3 Fig. Effect of IE2F depletion on TSR strength in HCMV Towne IE2F and TB40/E IE2F strains in late infection.** TSR strength determined for all viral TSRs. Repressed and activated TSRs increase and decrease in strength, respectively, when IE2F is depleted. Repressed TSRs containing a dTag / CTRL ratio greater than 2 are red. Activated TSRs with the ratio less than 0.5 are blue. TSRs not changing by this degree are indicated by a small black dot.
(TIF)

**S4 Fig. Examples of highly affected host genes. (A)** UCSC Genome Browser views of normalized PRO-Seq data for highly induced (CSF3, ISG15) and inhibited (TXNIP) genes upon HCMV TB40/E infection. GENCODE v31 basic annotations are displayed in black. TruQuant annotations are shown as black rectangles underneath GENCODE v31 annotations. **(B)** PRO-Seq data for largest changes upon IE2F depletion. BIK was selected as one of the most upregulated genes upon IE2 depletion. EID2B and CPED1 were identified as the most downregulated genes.
(TIF)

**S5 Fig. (A)** Map of HCMV Towne UL87 gene with carboxyl end fused to HA alone or HA plus FKBP12M (F). Positions of *Bam*HI sites depicted. **(B)** Agarose gel electrophoresis. *Bam*HI digestion of HCMV WT Towne BAC and recombinants GalK (placed at the C-terminus of UL87), vTBP$^H$, and vTBP$^H$F yields fragments of 3.48, 4.7, 3.5, and 3.8 kbp, respectively.
(TIF)

**S1 Table. Scaling factors for PRO-Seq datasets.**
(TIF)

**S2 Table. Oligonucleotides.**
(TIF)

**S1 Data. Viral transcription start-regions (TSRs), effect of IE2F depletion, and PFA sensitivity indexes (PSIs).** *TSR, denotes false-TSR.
(XLSX)

**S2 Data. Effects of infection and IE2F depletion on transcription of host genes.**
(XLSX)

## Acknowledgments

We thank Jay Bradner, George Winter, and Nathaneal Gray for kindly sharing reagents and their advice and Bodo Plachter for sharing his wisdom regarding UL83 RNA expression. We are grateful to Mark Stinski for his encouragement, expert advice, and critical review of the data.

## Author Contributions

**Conceptualization:** David H. Price, Jeffery L. Meier.

**Data curation:** Ming Li, Jeffery L. Meier.

**Formal analysis:** Ming Li, David H. Price, Jeffery L. Meier.

**Funding acquisition:** David H. Price, Jeffery L. Meier.

**Investigation:** Ming Li, Jeffery L. Meier.

**Methodology:** Ming Li, Christopher B. Ball, Geoffrey Collins, Qiaolin Hu, Jeffery L. Meier.

**Project administration:** David H. Price, Jeffery L. Meier.

**Resources:** David H. Price, Jeffery L. Meier.

**Software:** Christopher B. Ball, Geoffrey Collins, David H. Price.

**Supervision:** David H. Price, Jeffery L. Meier.

**Validation:** Donal S. Luse, David H. Price, Jeffery L. Meier.

**Visualization:** Ming Li, David H. Price, Jeffery L. Meier.

**Writing – original draft:** Ming Li, David H. Price, Jeffery L. Meier.

**Writing – review & editing:** Christopher B. Ball, Donal S. Luse, David H. Price.

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
