## [Decision Letter · Decision Letter 0]

2 Dec 2019

Dear Dr Meier,

Thank you very much for submitting your manuscript "Human cytomegalovirus IE2 drives transcription initiation from a distinct subtype of viral late promoters by host RNA polymerase II" (PPATHOGENS-D-19-01981) for review by PLOS Pathogens. Your manuscript was fully evaluated at the editorial level and by independent peer reviewers. The reviewers appreciated the attention to an important problem, but raised some concerns about the manuscript as it currently stands. In particular, reviewers noted that the mechanistic description of features associated with late gene transcription, including the phenotypes linked to vTBP depletion, were insufficiently explored. The following specific pints need to be addressed prior to resubmission:

1) Additional validation of the dTAG approach, described in point 4 by Reviewer 1.

2) Clarify whether IE2 is inhibiting the early vs. the late transcription mechanism, which is somewhat complicated by the exclusive use of PAA treatment, as described by reviewer 2.

3) A more comprehensive analysis of late gene transcription from the PRO-seq experiments performed when DNA replication was inhibited, as noted by Reviewer 3.

Please modify the manuscript according to the review recommendations, and address the specific points made by each reviewer.

(1) A letter containing a detailed list of your responses to the review comments and a description of the changes you have made in the manuscript. Please note while forming your response, if your article is accepted, you may have the opportunity to make the peer review history publicly available. The record will include editor decision letters (with reviews) and your responses to reviewer comments. If eligible, we will contact you to opt in or out.

(2) Two versions of the manuscript: one with either highlights or tracked changes denoting where the text has been changed; the other a clean version (uploaded as the manuscript file).

Additionally, to enhance the reproducibility of your results, PLOS recommends that you deposit your laboratory protocols in protocols.io, where a protocol can be assigned its own identifier (DOI) such that it can be cited independently in the future. For instructions see http://journals.plos.org/plospathogens/s/submission-guidelines#loc-materials-and-methods

We hope to receive your revised manuscript within 60 days. If you anticipate any delay in its return, we ask that you let us know the expected resubmission date by replying to this email. Revised manuscripts received beyond 60 days may require evaluation and peer review similar to that applied to newly submitted manuscripts.

[LINK]

Sincerely,

Britt A Glaunsinger

Guest Editor

PLOS Pathogens

Klaus Früh

Section Editor

PLOS Pathogens

Kasturi Haldar

Editor-in-Chief

PLOS Pathogens

orcid.org/0000-0001-5065-158X

Grant McFadden

Editor-in-Chief

PLOS Pathogens

orcid.org/0000-0002-2556-3526

Reviewer's Responses to Questions

**Part I - Summary**

Reviewer #1: This manuscript combines degradation tagging (dTAG) with precision nuclear run-on sequencing (PRO-Seq) to identify transcription start sites (TSS) that are dependent on the CMV IE2 protein(s). They first characterize the specificity of their approach – fusion of FKBP12_F36V to IE2 does not appear to impair CMV replication; however, in the presence of dTAG ligand (from 90-96 hpi) results in loss of all 3 IE2 isoforms (86, 60, 40), de-repression of the MIEP, but no effect on expression of the pp28 late gene product. They then use PRO-Seq, including addition of the P-TEFb inhibitor flavopiridol, to identify 18 TSS dependent on IE2 (Dependence was defined as 6h-dTAG treatment induced decrease in TSS read-number by >50% with TB40/E and >75% with Towne strain infections). Notably, 17 of the 18 IE2 dependent start sites were “highly PFA-sensitive.” They also examine host gene expression and find no substantial effect of IE2 at late times. Finally, they use the dTAG approach to degrade the UL87 encoded vTBP and show that this dramatically decreases expression of the canonical late gene encoded pp28 and pp71, there is minimal effect on UL83 encoded pp65. This, coupled with the lack of consistent TATT boxes upstream IE2 dependent TSS, leads them to assert that IE2 plays a role in transcription initiation from a distinct subtype of viral late promoters.

In summary, this manuscript is quite impressive. By using the dTAG system, the authors provide an elegant means of focusing on the role of IE2 proteins at late times post infection (despite its importance early). Coupled with the PRO-seq approach, this allowed them to identify a discreet set of IE2 dependent start sites. An important limitation of the manuscript is that determination of early vs. late kinetics relies on PFA treatment – a viral DNA synthesis inhibitor that abrogates late gene expression, but also impairs early transcription (compare read depths in Fig2C 72h vs. 72h PFA). This approach, long established and still widely used, has become more problematic with the realization that many promoters are transcribed both early and late (called variously early-late, leaky-late, gamma-1). Unless a promoter has an extremely strong early component (like UL83), the PFA method will classify most early-late promoters as late. This distinction is important because, if a significant proportion of the Table 1 TSS’s are actually early-late, the question arises whether IE2 is inhibiting the early vs. the late transcription mechanism. The UL83 gene presents the best opportunity to address this concern: if treatment with PFA + dTAG results in no further decrease in UL83 expression than PFA alone, this would firmly establish that IE2 is not effecting the early (i.e., the non-DNA replication dependent) contribution.

A final minor point is that the discussion of the different IE2 isoforms in introduction led me to expect some insights into the different roles of these isoforms. I agree with line 377 of the discussion and would consider placing less emphasis on the IE2 isoforms since the manuscript does not address this issue.

Reviewer #2: This is a study of the transcription of late HCMV genes, specifically the requirement for DNA replication, IE2, and cellular and viral GTFs that recognize the promoters. The authors use a tagging method to destabilize the IE2 proteins by the addition of the compound dTAG, PFA to inhibit viral DNA replication, and Pro-seq to measure transcription. What they conclude is summarized in the past two sentences of the discussion (lines 399-4023), “Our results support a model in which replicated HCMV genomes rid of repressive chromatin allow IE2 and LTFs to carry out their assigned function in selectively targeting viral late promoters for activation. IE2 and LTFs trans-activate late viral promoters by two distinct pathways - IE2 using an alpha-herpesvirus-like pathway and LTFs using a gamma-herpesvirus like pathway.”

It is interesting and quite clear that there are different types of late HCMV genes, with respect to their mechanism of transcription activation. The authors’ adaptation of Pro-seq is quite nice. The utilization of the dTAG to selectively reduce IE2 is laudable. However, the mechanism of late gene activation is not adequately explored beyond the the application of Pro-seq and inspection of promoter elements. There also needs to be a more rigorous validation of the dTAG approach in this case.

1. There are no experiments addressing repressive chromatin in this study, so statements like that above regarding chromatin are not supported.

2. The model diagrammed in Fig 6 is very vague and adds little to the paper.

3. The authors talk about the involvement of vTBP, TFIID, and LTF at select promoters, usually based on sequence analysis, but there are no studies specifically addressing the association of these factors at these specific promoters.

4. There are questions regarding the dTAG procedure.

a. How well does it approximate a IE2 null situation? There is some made when dTAG is added (fig S2). Therefore, if there are different quantitative requirements for IE2 for the activation of different genes, it is possible that this might be missed.

b. How does dTAG affect the life cycle of HCMV? A growth curve needs to be performed.

c. Likewise how does dTAG affect the replication kinetics of IE2F (S2A)? Additionally, the replication curve (S2A) only shows a 6.5 fold amplefication of HCMV DNA. Seems a bit low. Is this the whole life cycle?

5. Is there a Fig S3? I couldn’t find it.

Reviewer #3: In this study, the authors utilized a novel approach for targeted and timed degradation of the immediate-early IE2 protein isoforms encoded by CMV to investigate their role in viral transcription during late infection. They examined the impact of IE2 depletion and the inhibition of DNA replication on nascent transcriptomics by PRO-seq. Combining Pro-seq with flavopiridol they determined the transcription start sites and corresponding promoters. They found that most of the IE2-dependent transcribed viral promoters were also dependent on DNA replication and therefore are late gene promoters, except for the UL83 promoter. For beta/gamma-herpesviruses transcription of late genes is mediated by a viral preinitiation complex (vPIC) composed of six viral proteins including a viral TATA-binding protein (vTBP) that recognizes atypical TATT-containing sequences in the late gene promoters. However, the majority of the IE2-dependent promoters lacks a TATT sequence. Transcription of an IE2-dependent promoter, the UL83 promoter, was not affected by targeted degradation of vTBP. Taken together, the authors concluded that IE2 regulates the transcription of a subset of late promoters, many of which lack TATT and are likely independent of vPIC. This work supports previous studies in EBV by others that some of viral late genes are vPIC-independent. While this study indicates a role of IE2 in vPIC-independent late gene transcription, no mechanistic studies were attempted. Moreover, the authors did not elaborate on their Pro-seq studies on viral late gene transcription. For example, PRO-seq was performed when DNA replication was inhibited but a comprehensive analysis of viral late gene transcription was not included. A viral targeted degradation of vTBP recombinant was generated but was not used for Pro-seq to examine vPIC-independent late gene transcription.

**Part II – Major Issues: Key Experiments Required for Acceptance**

Reviewer #1: 1) The authors should assess whether UL83 (and pp65 protein) is further decreased by PFA + dTAG (using the hCMV IE2F virus) than PFA alone.

Reviewer #2: (No Response)

Reviewer #3: 1. PRO-seq was performed when DNA replication was inhibited. However, no analysis of viral gene transcription was provided. It is not clear that among late gene promoters identified through Pro-seq, how many lack TATT? How many of these TATT-less promoters depend on IE2? Pro-seq should be also carried out with targeted degradation of vTBP. These two Pro-seq datasets will provide a comprehensive description of vPIC-dependent as well as -independent viral late gene transcription in HCMV.

2. Transcription of UL83 is the only IE2-dependent gene examined in this study for the vTBP requirement. However, based on Table 1, transcription of the UL83 promoter is partially affected by the inhibition of DNA replication. Another IE2-dependent promoter that is highly sensitive to the inhibition of DNA replication should be examined in Fig. 6 Furthermore, three IE2-dependent promoters still have TATT. Does transcription of these three require vTBP?

**Part III – Minor Issues: Editorial and Data Presentation Modifications**

Reviewer #1: A supplementary table summarizing the data would be helpful. For each TSS detected (not just the 18 reported in Table 1) it should report:

1) A quantitative assessment of IE2 dependence (as a ratio or %) for both the TB40/E and Towne infections.

2) A quantitative assessment of PFA dependence.

3) For well studied genes, this table should be annotated with previously determined kinetics (early, early-late, or late).

This should provide a more complete picture of the effects of IE2 degradation at late times. It is somewhat surprising, given the role of IE2 in early promoter regulation, only late TSS sites were affected. It would be helpful if the authors could comment on this in the discussion (i.e., is the lack of an effect on early TSS’s surprising, why or why not?).

Reviewer #2: (No Response)

Reviewer #3: The Discussion of the manuscript was not written clearly and needs revisions. For example, it is difficult to understand what authors want to communicate in line 363-365 and in line 377-379.

PLOS authors have the option to publish the peer review history of their article (what does this mean?). If published, this will include your full peer review and any attached files.

Reviewer #1: No

Reviewer #2: No

Reviewer #3: No

---

## [Editor Report · Decision Letter 1]

13 Feb 2020

Dear Dr Meier,

We are pleased to inform you that your manuscript 'Human cytomegalovirus IE2 drives transcription initiation from a select subset of late infection viral promoters by host RNA polymerase II' has been provisionally accepted for publication in PLOS Pathogens.

Before your manuscript can be formally accepted you will need to complete some formatting changes, which you will receive in a follow up email. A member of our team will be in touch within two working days with a set of requests.

Best regards,

Britt A Glaunsinger

Guest Editor

PLOS Pathogens

Klaus Früh

Section Editor

PLOS Pathogens

Kasturi Haldar

Editor-in-Chief

PLOS Pathogens

orcid.org/0000-0001-5065-158X

Michael Malim

Editor-in-Chief

PLOS Pathogens

orcid.org/0000-0002-7699-2064
---

## [Editor Report · Acceptance letter]

30 Mar 2020

Dear Dr Meier,

We are delighted to inform you that your manuscript, "Human cytomegalovirus IE2 drives transcription initiation from a select subset of late infection viral promoters by host RNA polymerase II," has been formally accepted for publication in PLOS Pathogens.

Best regards,

Kasturi Haldar

Editor-in-Chief

PLOS Pathogens

orcid.org/0000-0001-5065-158X

Michael Malim

Editor-in-Chief

PLOS Pathogens

orcid.org/0000-0002-7699-2064